# Linear response and exact hydrodynamic projections in Lindblad equations with decoupled Bogoliubov hierarchies

Patrik Penc[1] and Fabian H.L. Essler[1]

[1] *The Rudolf Peierls Centre for Theoretical Physics,*
*Oxford University, Oxford OX1 3PU, United Kingdom*

(Dated: December 27, 2025)

We consider a class of spinless-fermion Lindblad equations that exhibit decoupled BBGKY hierarchies. In the cases where particle number is conserved, their late time behaviour is characterized by diffusive dynamics, leading to an infinite temperature steady state. Some of these models are Yang-Baxter integrable, others are not. The simple structure of the BBGKY hierarchy makes it possible to map the dynamics of Heisenberg-picture operators on few-body imaginary-time Schrödinger equations with non-Hermitian Hamiltonians. We use this formulation to obtain exact hydrodynamic projections of operators quadratic in fermions, and to determine linear response functions in Lindbladian non-equilibrium dynamics.

## CONTENTS

## I. INTRODUCTION

Understanding the effects of dissipation on the dynamics of many-particle systems is an important challenge. Significant simplifications occur in cases where the environmental degrees of freedom can be treated in a Markovian approximation, which allows a description in terms of a Lindblad equation (LE)[1–3]. Many-particle LEs are generally difficult to analyze and most studies are based on perturbative [4, 5] or numerical approaches [6–10]. This poses the question whether there are examples in which exact results can be obtained. One such class of models are LEs that can be written as imaginary-time Schrödinger equations with non-Hermitian "Hamiltonians" that are quadratic in fermionic or bosonic field op-

erators [11–20]. These are very special in that the density matrix retains its Gaussian form under time evolution. A wider and richer class of exactly solvable LEs was identified more recently with the discovery of Lindbladians that can be mapped to *interacting* Yang-Baxter integrable models [21–31]. Some models were found to exhibit operator-space fragmentation [32–34] – the Lindbladian analog of Hilbert-space fragmentation [35, 36]– which gives rise to novel kinds of Yang-Baxter integrability and free fermion structures respectively. In a separate development unrelated to integrability LEs that give rise to decoupled BBGKY hierarchies were identified [37–48], which hugely simplifies the analysis of correlation functions. Importantly, these models typically cannot be mapped onto LEs with Lindbladians that are quadratic in bosons or fermions, and do not preserve Gaussianity of density matrices under time evolution. Interestingly some LEs are both Yang-Baxter integrable and exhibit decoupled BBGKY hierarchies [21].

The purpose of our work is to exploit the simplifications afforded by decoupled BBGKY hierarchies to address three issues of current interest.

● First, we obtain analytic results for exact *hydrodynamic projections* [49]. We analyze LEs with a continuous U(1) symmetry related to particle number conservation and infinite temperature steady states, which exhibit diffusive behaviour at late times. The presence of a conservation law then implies that certain observables will exhibit hydrodynamic power-law tails at late times, which are related to the existence of diffusive eigenmodes of the Lindbladian. A generally unsolved problem is to determine the projection of a given operator onto these diffusive modes. Generically only symmetry arguments are available to gain some information on such projections, see e.g. [50] for a recent example. In order to put some of our results into context we briefly review the usual arguments pertaining to diffusive late-time dynamics and its relation to conservation laws. Let us consider a lattice model with U(1) symmetry corresponding to particle number conservation. The particle density $n_m$ then fulfils a continuity equation

$$\frac{d}{dt}n_m(t) = j_{m-1}(t) - j_m(t) , \qquad (1)$$

where $j_m$ is the particle current density at site $m$. Coarse-graining turns this into

$$\frac{\partial}{\partial t}\rho(t,x) = -\frac{\partial}{\partial x}J(t,x) . \qquad (2)$$

Using that the expectation value of the current (in a spatially inhomogeneous state) is related to the gradient of the density

$$\langle J(t,x)\rangle = -D\frac{\partial}{\partial x}\langle\rho(t,x)\rangle + \ldots \qquad (3)$$

then gives a diffusion equation for $\langle\rho(t,x)\rangle$. Its solution can be cast in the form

$$\langle\rho(t>0,x)\rangle = \int\frac{dk}{2\pi}e^{-Dk^2 t}\int_{-\infty}^{\infty}dx'e^{ik(x-x')}\langle\rho(0,x')\rangle . \qquad (4)$$

This implies that the Heisenberg-picture operator at late times has hydrodynamic tails

$$\rho(t\to\infty,x) \sim \frac{A(x)}{\sqrt{t}} + \ldots \qquad (5)$$

In practice one often is not interested in coarse grained quantities but rather microscopic ones. The question is then how the existence of slow coarse-grained operators translates to the late-time behaviour of lattice operators, e.g. $(n_m n_{m+1})(t)$. This "hydrodynamic projection" is a difficult problem whose solution is generally not know. Eqn (4) further implies that the (super)operator $\mathcal{L}[.]$ generating time evolution

$$\frac{d}{dt}\hat{X} = \mathcal{L}[\hat{X}] \qquad (6)$$

must have eigenvalues in the momentum representation that behave for small momenta like $\lambda(k) = -Dk^2 + \ldots$. The corresponding eigenoperators are referred to as *diffusive eigenmodes* of the time evolution operator. Their construction is another difficult problem whose solution is not known. In our work we derive explicit expressions for them.

● Second, we consider linear response functions on top of Lindbladian time evolution [51]. This is relevant in the context of pump-probe experiments, and in particular the question how dissipation affects the line shapes seen in linear response functions. Analyzing linear response on top of Lindbladian time evolution is generally a difficult problem, but as we show in the following, this becomes feasible in a class of models with decoupled BBGKY hierarchies.

● Third, we utlize the fact that some LEs with decoupled BBGKY hierarchies are Yang-Baxter integrable while others are not to investigate what effects integrability has. In contrast to unitary quench dynamics [52] integrability in LEs does not imply the existence of conservation laws. This is immediately obvious from the fact that many integrable LEs have unique steady states that are completely mixed, i.e. correspond to infinite temperature density matrices.

The outline of this manuscript is as follows. In section II we introduce the class of LE studied in our work. Section IV shows that by an appropriate vectorization the equations of motion in the Heisenberg picture can be mapped onto imaginary time Schrödinger equations with non-Hermitian Hamiltonians. The latter take the form of spin-1/2 fermions with purely imaginary hopping terms and short-ranged interactions. Section V focuses on the time evolution of operators quadratic in fermions in U(1) symmetric models and presents a detailed analysis of the corresponding eigenvalue spectrum of the non-Hermitian Hamiltonians. In section VI closed-form expressions for

the hydrodynamic projections of U(1) invariant operators are derived. These describe the late-time behaviour of such operators. Section VII discusses the time evolution in LEs with decoupled BBGKY hierarchies that break particle-number conservation. Here the decoupling results in a triangular form of the BBGKY hierarchy, and we show how to solve the resulting equations of motion for operators quadratic in fermions. Operators cubic in fermions decay exponentially in time in all LEs studied here, and Section VIII reports the corresponding decay rates. In contrast, operators quartic in fermions can again display hydrodynamic tails. Section IX discusses the spectra of the non-Hermitian Hamiltonians relevant to the late time behaviour, focusing in particular on a Yang-Baxter integrable case in which an analytic understanding is possible. In sections X and XI we apply our results to the study of non-equal time correlation functions in the steady state and linear response functions after quantum quenches respectively. Finally, we summarize our results and discuss lines of future enquiry in Section XII.

## II. LINDBLAD EQUATIONS WITH DECOUPLED BOGOLIUBOV HIERARCHIES

The evolution equation for the reduced density matrix $\rho(t)$ in an open quantum system described by a Lindblad equation reads

$$\mathcal{L}[\rho(t)] = -i\left[H,\ \rho(t)\right] + \sum_\alpha 2L_\alpha \rho(t) L_\alpha^\dagger - \left\{L_\alpha^\dagger L_\alpha,\ \rho(t)\right\},$$

$$\partial_t \rho(t) = \mathcal{L}[\rho(t)]\ , \tag{7}$$

Here $H$ is the effective Hamiltonian of the system and $L_\alpha$ are the jump operators characterising the interaction between the system and the environment. For our purposes it is convenient to work in the Heisenberg picture, where time evolution of operators is governed by the dual Lindbladian

$$\mathcal{L}^*[A(t)] = i\left[H,\ A(t)\right] + \sum_\alpha 2L_\alpha^\dagger A(t) L_\alpha - \left\{L_\alpha^\dagger L_\alpha,\ A(t)\right\}\ . \tag{8}$$

If the set $\{L_\alpha\}$ of jump operators is closed under Hermitian conjugation the Lindblad equation takes the simpler form

$$\mathcal{L}^*[A(t)] = i\left[H,\ A(t)\right] - \sum_\alpha \left[L_\alpha^\dagger,\ [L_\alpha,\ A(t)]\right]\ . \tag{9}$$

From hereon we focus on many-particle system of spinless fermions

$$\{c_j, c_\ell\} = 0\ ,\quad \{c_j, c_\ell^\dagger\} = \delta_{j,\ell}\ . \tag{10}$$

The equations of motions for products of fermion operators then generically give rise to a generalization of the celebrated Bogoliubov–Born–Green–Kirkwood–Yvon (BBGKY) hierarchy [53]. However, if we choose both the Hamiltonian $H$ and all jump operators $L_\alpha$ to be quadratic in fermion operators, the hierarchy decouples as a result of the double-commutator structure of the Lindbladian (9), and one obtains *closed* systems of equations for operators involving at most $n$ fermion operators. It is worthwhile to stress the following points:

- Such models are not free (there is no Wick's theorem) and an intially Gaussian density matrix does not remain Gaussian under time evolution.

- The decoupling of the BBGKY hierarchy is possible in any number of spatial dimensions.

These observations have already been exploited to obtain exact results for equal-time observables in some such models [30, 31, 54]. We go beyond these works here in order to study linear response functions and obtain an exact description of the hydrodynamic behaviour that arises at late times.

We choose the Hamiltonian part to be a one-dimensional tight-binding model with periodic boundary conditions

$$H = -J\sum_{j=1}^{L}\left(c_{j+1}^\dagger c_j + c_j^\dagger c_{j+1}\right)\ . \tag{11}$$

The Hamiltonian preserves fermion number and has a global $U(1)$ symmetry

$$c_j \to c_j e^{i\alpha}\ . \tag{12}$$

We consider the following choices of jump operators (all of which lead to decoupled BBGKY hierarchies)

I. On-site dephasing noise .

$$L_j = \sqrt{\gamma} c_j^\dagger c_j\ . \tag{13}$$

The corresponding Lindblad equation maintains the U(1) symmetry and is known to be Yang-Baxter integrable [21]. More precisely it can be mapped to the Hubbard model with imaginary interaction, which will be useful in the following.

II. "Two-channel" dephasing noise acting across a bond.

$$L_{j,1} = \sqrt{\frac{\gamma}{2}} c_j^\dagger c_{j+1}\ ,\quad L_{j,2} = \sqrt{\frac{\gamma}{2}} c_{j+1}^\dagger c_j\ . \tag{14}$$

This is characterized by two dissipation channels with equal rates. The resulting Lindblad equation is non-integrable and generalizes the Quantum Simple Symmetric Exclusion Process [32, 55–59] by introducing a Hamiltonian part.

III. Single channel dephasing noise acting across a bond. This is an elaboration of Model I. We consider the following jump operators:

$$L_j = \sqrt{\frac{\gamma}{2}}\left(n_j + \nu\ n_{j+1}\right)\ ,\quad \nu \in \{\pm 1\}. \tag{15}$$

IV. Single channel dephasing noise acting across a bond. This is an elaboration of Model II. Here we consider the following two choices:

$$\text{(a)} \quad L_j = \sqrt{\frac{\gamma}{2}} \left( c_j^\dagger c_{j+1} + c_{j+1}^\dagger c_j \right) \ ,$$

$$\text{(b)} \quad L_j = i\sqrt{\frac{\gamma}{2}} \left( c_j^\dagger c_{j+1} - c_{j+1}^\dagger c_j \right) \ . \quad (16)$$

V. Finally we consider jump operators that break the $U(1)$ symmetry:

$$L_{j,1} = \sqrt{\frac{\gamma}{2}} c_j c_{j+1} \ , \quad L_{j,2} = \sqrt{\frac{\gamma}{2}} c_{j+1}^\dagger c_j^\dagger \ . \quad (17)$$

All the models have the infinite temperature (equilibrium) state as their steady state. Models III and IV are chosen in order to give a clear picture of how the spatial structure of the jump operators affects the properties of the hydrodynamic modes, without making the analysis overly tedious.

## III.   OVERVIEW OF MAIN RESULTS

In order make the manuscript easier to read we now provide a brief overview of the main results derived in the remainder of the manuscript.

### A.   Exact expressions for diffusive eigenoperators

Models I-IV have eigenoperators of the dual Lindbladian that exhibit diffusive behaviour at small momentum

$$\mathcal{L}^*[\hat{\Phi}(p)] = \lambda(p)\hat{\Phi}(p) \ , \quad \lambda(p \to 0) = -Dp^2 + \dots \ . \quad (18)$$

We derive explicit expressions for these eigenoperators in the sector with one fermion creation and annihilation operator each. These have the form

$$\hat{\Phi}(p) = \sum_{j,\ell} e^{ip\frac{j+\ell}{2}} \phi(p; j - \ell) c_j^\dagger c_\ell \ , \quad (19)$$

where the amplitudes $\phi(p; j - \ell)$ are model-dependent. In all cases they describe "particle-hole bound states" in the sense that $\phi(p; j - \ell)$ decay exponentially with respect to the distance $|j - \ell|$. The corresponding decay length, or size of the bound state, tends to zero in the zero momentum limit $p \to 0$.

### B.   Exact hydrodynamic projections

The existence of diffusive eigenoperators implies the presence of long-time tails in the expressions of certain Heisenberg-picture operators. We derive explicit expressions for these hydrodynamic projections. In particular we show that

$$(c_{x_0}^\dagger c_{y_0})(t \gg \gamma^{-1}) \simeq \sum_{x,y} \psi_{x_0,y_0}(x,y;t) c_x^\dagger c_y \ , \quad (20)$$

where the amplitudes $\psi_{x_0,y_0}(x,y;t)$ are model-dependent, but are all expressed in terms on confluent hypergeometric functions, $cf.$ (109). At asymptotically late times they reduce to (decaying) power-laws in $t$, with exponents that depend on $X = |x - y| + |x_0 - y_0|$ and $Y = (x - x_0) + (y - y_0)$

$$\psi_{x_0,y_0}(x,y;t)\Big|_{\text{leading}} = \begin{cases} A_e \ (Dt)^{-\frac{1+X}{2}} & \text{Y even} \ , \\ A_o \ (Dt)^{-\frac{2+X}{2}} & \text{Y odd} \ . \end{cases} \quad (21)$$

Here $D$ is model-dependent.

### C.   Non-equal time steady-state correlations

The steady state of models I-IV is a simple infinite-temperature state. We show that equilibrium dynamics in this state can be efficiently analyzed. In particular, signatures of the hydrodynamic modes can be seen in dynamical correlation functions. We show this for the case of the Fourier transform of the density-density correlation function

$$G(q,\omega) = \sum_q \int_0^\infty \mathrm{d}t \ e^{i(\omega t - qx)} \left[ \frac{1}{2^L} \mathrm{Tr}\left[n_j(t)n_\ell\right] - \frac{1}{4} \right]$$

$$\sim \frac{A}{i\omega + Dq^2} \ , \quad \omega, q \to 0. \quad (22)$$

### D.   Linear response functions in non-equilibrium states

Linear response on top of non-equilibrium time evolution describes certain "pump-probe" experiments. We analyze this setting for Lindblad equations with decoupled BBGKY hierarchies. In particular we consider the case of the density response to an infinitesimal density perturbation, described by the Lindblad equation

$$\frac{d\rho(t)}{dt} = \mathcal{L}_t[\rho(t)] = \mathcal{L}_0[\rho(t)] - i\sum_j \xi_j(t)[n_j, \rho(t)] \ . \quad (23)$$

The linear response is given by the term linear in $\xi_j(t)$

$$\mathrm{Tr}[n_j\rho(t)] - \mathrm{Tr}[n_j\rho_0(t)] = \int_0^\infty dt' \sum_j \xi_j(t')\chi(j, \ell; t, t'),$$

$$\quad (24)$$

where $\rho_0(t)$ in density matrix in absence of the perturbation. We determine the Fourier transform of the susceptibility $\chi(j, \ell; t, t')$ for initial density matrices corresponding to the ground state of dimerized tight-binding models and show how dissipation affects the dynamical response.

## IV. STRUCTURE OF THE EQUATIONS OF MOTION

Let us consider the following set of operators

$$\mathcal{O}(\vec{j}_n, \vec{\ell}_m) = c_{j_1}^\dagger \dots c_{j_n}^\dagger c_{\ell_1} \dots c_{\ell_m} \ ,$$
$$\hat{\Psi}_{n,m} = \sum_{\vec{j}_n, \vec{\ell}_m} \Psi(\vec{j}_n, \vec{\ell}_m)\mathcal{O}(\vec{j}_n, \vec{\ell}_m) \ . \qquad (25)$$

In order to derive the equations of motion of $\hat{\Psi}_{n,m}$ we vectorize

$$\mathbb{1}_j \longrightarrow |0\rangle_j \ ,$$
$$c_j^\dagger \longrightarrow c_{j,\uparrow}^\dagger |0\rangle_j \ ,$$
$$c_j \longrightarrow (-1)^j c_{j,\downarrow}^\dagger |0\rangle_j \ ,$$
$$c_j^\dagger c_j \longrightarrow (-1)^j c_{j,\uparrow}^\dagger c_{j,\downarrow}^\dagger |0\rangle_j \ . \qquad (26)$$

The equations of motion then take the form of an imaginary time Schrödinger equation with a non-Hermitian Hamiltonian for the vectorized operator $\hat{\Psi}_{n,m}$

$$\frac{d}{dt}|\Psi_{n,m}(t)\rangle = \mathcal{H}|\Psi_{n,m}(t)\rangle \ ,$$
$$|\Psi_{n,m}(0)\rangle = \sum_{\vec{j}_n, \vec{\ell}_m} \Psi(\vec{j}_n, \vec{\ell}_m) \prod_{r=1}^n c_{j_r,\uparrow}^\dagger \prod_{s=1}^m c_{\ell_s,\downarrow}^\dagger |0\rangle$$
$$\times (-1)^{\sum_{s=1}^m \ell_s}. \qquad (27)$$

The factors of $(-1)^{\ell_s}$ have been introduced in order for the Hamiltonians to have the same hopping terms for spin-up and spin-down fermions. It is important to note that this leads to a shift in total momentum by $\pi$ in the vectorized formalism compared to the underlying equation of motion for the operators $\mathcal{O}(\vec{j}, \vec{\ell})$.

The various choices of jump operators above give rise to the following Hamiltonians:

- Model I:

$$\mathcal{H} = -iJ \sum_{j,\sigma} \left(c_{j,\sigma}^\dagger c_{j+1,\sigma} + \text{h.c.}\right)$$
$$+ 2\gamma \sum_j \left[(n_{j,\uparrow} - \frac{1}{2})(n_{j,\downarrow} - \frac{1}{2}) - \frac{1}{4}\right]. \qquad (28)$$

This is the one-dimensional Hubbard model with imaginary hopping term. In this case the equivalence of the equations of motion for $n$-particle Green's functions and wave functions of the Hubbard model was first observed in Ref. [21].

- Model II:

$$\mathcal{H} = -iJ \sum_{j,\sigma} [c_{j,\sigma}^\dagger c_{j+1,\sigma} + \text{h.c.}] - \gamma \sum_j [P_{j+1}^\dagger P_j + \text{h.c.}]$$
$$+ \gamma \sum_{j,\sigma} \left[(n_{j,\sigma} - \frac{1}{2})(n_{j+1,\sigma} - \frac{1}{2}) - \frac{1}{4}\right], \qquad (29)$$

where we have defined a pair annihilation operator $P_j = c_{j,\downarrow}c_{j,\uparrow}$.

- Model III:

$$\mathcal{H} = -iJ \sum_{j,\tau} \left(c_{j,\tau}^\dagger c_{j+1,\tau} + \text{h.c.}\right) - \nu\gamma \sum_j S_j^z S_{j+1}^z$$
$$+ 2\gamma \sum_j \left[(n_{j,\uparrow} - \frac{1}{2})(n_{j,\downarrow} - \frac{1}{2}) - \frac{1}{4}\right], \qquad (30)$$

where $S_j^z = n_{j,\uparrow} - n_{j,\downarrow}$ and we have used that $\nu^2 = 1$.

- Models IV:

$$\mathcal{H} = -iJ \sum_{j,\sigma} [c_{j,\sigma}^\dagger c_{j+1,\sigma} + \text{h.c.}]$$
$$+ \gamma \sum_{j,\sigma} \left[(n_{j,\sigma} - \frac{1}{2})(n_{j+1,\sigma} - \frac{1}{2}) - \frac{1}{4}\right]$$
$$- \gamma \sum_j [P_{j+1}^\dagger P_j - \xi S_{j+1}^+ S_j^- + \text{h.c.}], \qquad (31)$$

where $S_j^- = c_{j,\downarrow}^\dagger c_{j,\uparrow}$ are spin lowering operators. We have $\xi = \pm 1$ for models IV(a) and IV(b) respectively, while setting $\xi = 0$ recovers Model II.

- Model V:

$$\mathcal{H} = -iJ \sum_{j,\tau} \left(c_{j,\tau}^\dagger c_{j+1,\tau} + \text{h.c.}\right)$$
$$+ \gamma \sum_{j,\sigma} \left[(n_{j,\sigma} - \frac{1}{2})(n_{j+1,-\sigma} - \frac{1}{2}) - \frac{1}{4}\right]$$
$$+ \gamma \sum_j (P_j^\dagger P_{j+1} + P_{j+1}^\dagger P_j - P_j P_{j+1})$$
$$+ \gamma \sum_j (-1)^j \left[(n_j - 1)P_{j+1} - P_j(n_{j+1} - 1)\right] \ , \qquad (32)$$

where we have defined $n_j = n_{j,\uparrow} + n_{j,\downarrow}$. A key feature of this Hamiltonian is that in all terms there are at least as many fermion annihilation operators as there are creation operators.

### A. Formal solution of the equations of motion

In order to solve the equations of motion for the operators of interest we proceed along the same lines as in few-particle Quantum Mechanics: we first solve the time-independet Schrödinger equation for the right eigenstates

$$\mathcal{H}|\Phi_{R,\alpha}\rangle = \lambda_\alpha |\Phi_{R,\alpha}\rangle \ , \qquad (33)$$

and then obtain the solution of the time-dependent problem as

$$|\Psi_{n,m}(t)\rangle = \sum_\alpha \langle \Phi_{L,\alpha}|\Psi_{n,m}(0)\rangle e^{\lambda_\alpha t} |\Phi_{R,\alpha}\rangle \ . \qquad (34)$$

Here $\langle \Phi_{L,\alpha}|$ are left eigenstates with eigenvalue $\lambda_\alpha$, normalized such that

$$\langle \Phi_{L,\alpha}|\Phi_{R,\beta}\rangle = \delta_{\alpha,\beta} \ . \qquad (35)$$

The eigenvalues $\lambda_\alpha$ have non-positive real parts and at sufficiently late times the decay of the operator will therefore be governed by the eigenvalue whose real part is closest to zero. We call

$$\Delta_{n,m} = \max_\alpha \{\text{Re}(\lambda_\alpha)\} \qquad (36)$$

the *decay rate* of the operator $\hat{\Psi}_{n,m}$. The structure of the right eigenvalue equation in models I-IV is simple because

1. The vaccum state $|0\rangle$ defined by $c_{j,\sigma}|0\rangle = 0$ is an exact eigenstate with eigenvalue zero;

2. The Hamiltonians commute with the number operators of spin up and spin down fermions

$$[\mathcal{H}, N_\sigma] = 0 \ , \quad N_\sigma = \sum_j n_{j,\sigma} \ . \qquad (37)$$

This means we can consider eigenstates in sectors with fixed numbers $N_\sigma$ of fermions with spin $\sigma$, i.e.

$$|\Phi_{R,\alpha}\rangle = \sum_{\vec{j}_{N_\uparrow}, \vec{\ell}_{N_\downarrow}} \Phi_\alpha(\vec{j}_{N_\uparrow}, \vec{\ell}_{N_\downarrow}) \prod_{r=1}^{N_\uparrow} c_{j_r,\uparrow}^\dagger \prod_{s=1}^{N_\downarrow} c_{\ell_s,\downarrow}^\dagger |0\rangle. \qquad (38)$$

In model V the fermion vacuum $|0\rangle$ is still an exact eigenstate, and while particle number is no longer conserved the structure of the Hamiltonian implies that eigenstates have the form

$$|\Phi_{R,\alpha}\rangle = \sum_{\substack{n \le N_\uparrow \\ m \le N_\downarrow}} \sum_{\vec{j}_n, \vec{\ell}_m} \Phi_\alpha^{(n,m)}(\vec{j}_n, \vec{\ell}_m) \prod_{r=1}^{n} c_{j_r,\uparrow}^\dagger \prod_{s=1}^{m} c_{\ell_s,\downarrow}^\dagger |0\rangle \ , \qquad (39)$$

where in the sums over $n$ and $m$ the difference $n - m = N_\uparrow - N_\downarrow$ is kept fixed.

### B. Single-fermion operators

For all models considered here the momentum space creation and annihilation operators

$$c(q) = \frac{1}{\sqrt{L}} \sum_j e^{iqj} c_j \ , \ q = \frac{2\pi n}{L} \ , \ 0 \le n < L \ , \qquad (40)$$

are eigenoperators of the dual Lindbladian. The Heisenberg picture operators take the simple form

$$c(k;t) = e^{(2iJ\cos(k)-\gamma)t} c(k) \ . \qquad (41)$$

The decay rates are momentum independent

$$\Delta_{0,1} = -\gamma \ . \qquad (42)$$

## V. DYNAMICS OF OPERATORS QUADRATIC IN FERMIONS IN $U(1)$-SYMMETRIC MODELS

We next turn to the dynamics of operators quadratic in fermions. It is convenient to treat Model V separately froms models I-IV, as particle number is not conserved. We recapitulate the steps set out above:

1. We consider operators of the form (34) with $n+m = 2$, e.g.

$$\hat{\Psi}_{1,1} = \sum_{j,\ell} \Psi(j,\ell) c_j^\dagger c_\ell \ . \qquad (43)$$

2. The vectorized form of the equation of motion in the Heisenberg picture then becomes, *cf.* (27)

$$\frac{d}{dt}|\Psi_{1,1}(t)\rangle = \mathcal{H}|\Psi_{1,1}(t)\rangle \ . \qquad (44)$$

This is a 2-particle imaginary time Schrödinger equation with non-Hermitian Hamiltonian.

3. This is solved by expanding $|\Psi_{1,1}(t)\rangle$ in a basis of two-particle right eigenstates of $\mathcal{H}$, see eqn (34). We therefore require the two-particle right eigenstates of $\mathcal{H}$

$$\mathcal{H}|\Phi_{R,\alpha}\rangle = \lambda_\alpha |\Phi_{R,\alpha}\rangle. \qquad (45)$$

These are most easily constructed in the position representation, which we discuss in detail in the following. Some of the corresponding steps in the momentum representation are briefly summarized in Appendix A.

### A. Position representation for the eigenvalue equation with $N_\uparrow = N_\downarrow = 1$

The position representation of the eigenvalue equation (33) reads

$$\sum_{x',y'} \mathfrak{H}(x,y|x',y') \Phi_{R,\alpha}(x',y') = \lambda_\alpha \Phi_{R,\alpha}(x,y) \ , \qquad (46)$$

where for Model I and III

$$\begin{aligned} \mathfrak{H}(x,y|x',y') = &-iJ \sum_{\tau=\pm 1} (\delta_{x,x'+\tau}\delta_{y,y'} + \delta_{x,x'}\delta_{y,y'+\tau}) \\ &- 2\gamma (1 - \delta_{x,y}) \delta_{x,x'}\delta_{y,y'} \\ &+ \nu \sum_{\tau=\pm 1} \delta_{x,y+\tau}\delta_{x,x'}\delta_{y,y'} \ , \end{aligned} \qquad (47)$$

while for Models II and IV we have

$$\begin{aligned} \mathfrak{H}(x,y|x',y') = &-iJ \sum_{\tau=\pm 1} (\delta_{x,x'+\tau}\delta_{y,y'} + \delta_{x,x'}\delta_{y,y'+\tau}) \\ &- \gamma \sum_{\tau=\pm 1} (\delta_{x,y}\delta_{x,x'+\tau}\delta_{y,y'+\tau} + \xi\delta_{x,y+\tau}\delta_{x,y'}\delta_{y,x'}) \\ &- 2\gamma \delta_{x,x'}\delta_{y,y'} \ . \end{aligned} \qquad (48)$$

The eigenvalue equation for Model I is identical to the eigenvalue equation in the one dimensional Hubbard model with imaginary hopping in the sector with one spin-up and one spin-down fermion [60], *cf.* Ref. [21]. We can construct eigenfunctions of $\mathfrak{H}$ for Models I-IV by a plane-wave ansatz

$$\Phi_{R,\alpha}(x,y) = \begin{cases} Ae^{i(k_1 x + k_2 y)} + Be^{i(k_2 x + k_1 y)} & \text{if } x > y, \\ T(x) & \text{if } x = y, \\ Ce^{i(k_1 x + k_2 y)} + De^{i(k_2 x + k_1 y)} & \text{if } x < y. \end{cases} \tag{49}$$

Solving the Schrödinger equation for $|x-y| \gg 1$ fixes the eigenvalue in terms of the wave numbers $k_{1,2}$

$$\lambda_\alpha = -2\gamma - 2iJ(\cos(k_1) + \cos(k_2)) . \tag{50}$$

The coefficients $A$, $B$, $C$, $D$ and $T(x)$ are determined by the combination of the eigenvalue equation, overall normalization and the boundary conditions

$$\Phi_\alpha(L+1,y) = \Phi_\alpha(1,y) , \ \Phi_\alpha(x,L+1) = \Phi_\alpha(x,1) . \tag{51}$$

Eqns (51) give

$$C = Ae^{-ik_2 L} , \quad D = Be^{-ik_1 L} , \tag{52}$$
$$1 = e^{i(k_1+k_2)L} . \tag{53}$$

The last equation quantizes the centre-of-mass momentum

$$k_{1,2} = \frac{p_n}{2} \pm q_n , \ p_n = \frac{2\pi n}{L} , \ n = 1, \dots, L . \tag{54}$$

The remaining equations fix the ratio $B/A$ and quantize the momentum $q_n$ of the relative motion. We find

1. Model I: Either $B = -A$ and

$$k_{1,2} = \frac{2\pi n_{1,2}}{L} , \ -\frac{L}{2} \le n_1 < n_2 < \frac{L}{2} , \tag{55}$$

or $B = Ae^{ik_1 L}$ and

$$(-1)^n e^{iq_n L} = \frac{2J\cos(p_n/2)\sin(q_n) + \gamma}{2J\cos(p_n/2)\sin(q_n) - \gamma} . \tag{56}$$

We note that the first class of solutions corresponds to SU(2) descendants of spin highest-weight states of the Hubbard model [61, 62].

2. Model II: Either $B = -A$ and

$$k_{1,2} = \frac{2\pi n_{1,2}}{L} , \ -\frac{L}{2} \le n_1 < n_2 < \frac{L}{2} , \tag{57}$$

or $B = Ae^{ik_1 L}$ and

$$(-1)^n e^{iq_n L} = \frac{2J\cos(p_n/2)\sin(q_n) - \gamma\cos(p_n)}{2J\cos(p_n/2)\sin(q_n) + \gamma\cos(p_n)} . \tag{58}$$

3. Model III: Either $B = -Ae^{ik_1 L}$ and

$$(-1)^n e^{iq_n L} = \frac{2J\cos(p_n/2) - i\nu\gamma e^{iq_n}}{2J\cos(p_n/2) - i\nu\gamma e^{-iq_n}} \tag{59}$$

or $B = Ae^{ik_1 L}$ and

$$(-1)^n e^{iq_n L} = \frac{2J\cos(p_n/2)(2J\cos(p_n/2)\sin(q_n) + \gamma) + \nu\gamma e^{iq_n}(2J\cos(p_n/2)\cos(q_n) - i\gamma)}{2J\cos(p_n/2)(2J\cos(p_n/2)\sin(q_n) - \gamma) - \nu\gamma e^{-iq_n}(2J\cos(p_n/2)\cos(q_n) - i\gamma)} . \tag{60}$$

4. Model IV: Either $B = -Ae^{ik_1 L}$ and

$$(-1)^n e^{iq_n L} = \frac{2J\cos(p_n/2) - i\xi\gamma e^{iq_n}}{2J\cos(p_n/2) - i\xi\gamma e^{-iq_n}} \tag{61}$$

or $B = Ae^{ik_1 L}$ and

$$(-1)^n e^{iq_n L} = \frac{2J\cos(p_n/2)(2J\cos(p_n/2)\sin(q_n) - \gamma\cos(p_n)) - \xi\gamma e^{iq_n}(2J\cos(p_n/2)\cos(q_n) + i\gamma\cos(p_n))}{2J\cos(p_n/2)(2J\cos(p_n/2)\sin(q_n) + \gamma\cos(p_n)) + \xi\gamma e^{-iq_n}(2J\cos(p_n/2)\cos(q_n) + i\gamma\cos(p_n))} . \tag{62}$$

### 1. Eigenvalue spectrum for Model I

The (finite size) spectrum of eigenvalues is easily computed numerically. Given that we are dealing with translationally invariant systems we can label the eigenvalues by their total momentum $p_n$. Results for $L = 100$ and $\gamma = 3$ are shown in Figs 1 and 2. We see that there is a continuum of eigenvalues with non-vanishing imaginary parts and real parts close to $-2\gamma$, *cf.* Fig. 1. In addition there is a band of real eigenvalues that tends

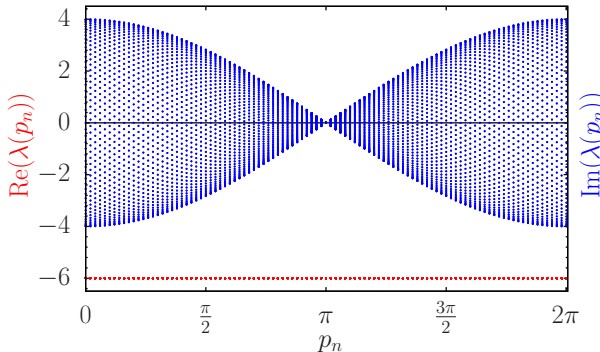

FIG. 1. Complex eigenvalues as a function of total momentum for Model I with $\gamma = 3$, $J = 1$ and $L = 100$. The real parts are approximately $-2\gamma$, indicating that the corresponding modes are strongly damped.

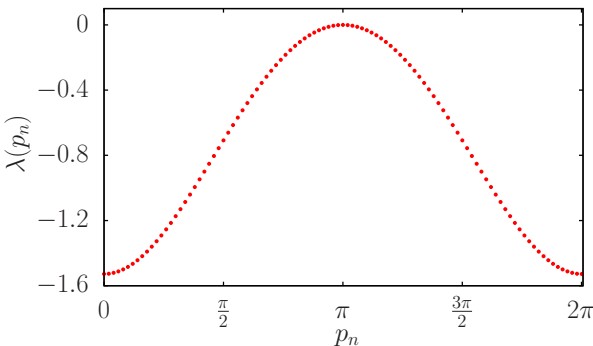

FIG. 2. Real eigenvalues as a function of total momentum for Model I with $\gamma = 3$, $J = 1$ and $L = 100$. The eigenvalues vanish as $(p_n - \pi)^2$ for $p_n \to \pi$, reflecting the existence of a diffusive hydrodynamic mode.

to zero as $(p_n - \pi)^2$ for total momentum $p_n \approx \pi$. The mode occurs around $p_n = \pi$ in our two-particle problem because of the alternating sign introduced in the vectorization (26) of the annihilation operators $c_j$. This leads to a shift in total momentum by $\pi$ in the vectorized formalism compared to the underlying equation of motion for the operators $\mathcal{O}(\vec{j}, \vec{\ell})$. In the original (unvectorized) problem the mode therefore corresponds to a family of eigenoperators of the time evolution of the form

$$\hat{\Phi}(P) = \sum_{x,y} \Phi_{R,P+\pi}(x,y)(-1)^y c_x^\dagger c_y \ , \qquad (63)$$

which have eigenvalues

$$\lambda(P \to 0) = -\frac{J^2}{\gamma} P^2 + \dots . \qquad (64)$$

For small momenta these are diffusive hydrodynamic modes related to the particle number $U(1)$ symmetry of the model, see the discussion in the introduction. Both types of solutions shown in Figs 1 and 2 are easily understood in the infinite volume limit, in which both $p_n$

and $q_n$ turn into continuous variables. The two classes of solutions shown in Figs 1 and 2 respectively then give rise to eigenvalues of the following form:

- $q \in \mathbb{R}$ and $\lambda(p,q) = -2\gamma - 4iJ \cos(\frac{p}{2}) \cos(q)$ This describes a two-parameter continuum of eigenstates, *cf.* Fig. 1.

- $q = \kappa + i\eta \in \mathbb{C}$ with $\kappa = -\frac{\pi}{2}\mathrm{sgn}\left(J\cos(\frac{p}{2})\right)$ and $\eta = \mathrm{arccosh}\left(\left|\frac{\gamma}{2J\cos(p/2)}\right|\right)$. This solution exists if $\left|\frac{2J\cos(p/2)}{\gamma}\right| < 1$ and gives rise to a single momentum-dependent mode with real eigenvalues

$$\lambda(p) = -2\gamma\left(1 - \sqrt{1 - \left(\frac{2J\cos(p/2)}{\gamma}\right)^2}\right). \qquad (65)$$

In a finite volume $L$ the $q_n$'s are quantized via the Bethe equations (56). However, for large $L$ the particular solutions of interest here can be obtained as (*cf.* [60, 63])

$$q_n = \kappa_n + i\eta_n + \mathcal{O}(e^{-\delta L}) \ , \quad \delta > 0 \ ,$$
$$\kappa_n = -\frac{\pi}{2}\mathrm{sgn}\left(J\cos(\frac{p_n}{2})\right) \ ,$$
$$\eta_n = \mathrm{arccosh}\left(\left|\frac{\gamma}{2J\cos(p_n/2)}\right|\right), \qquad (66)$$

where we recall that $p_n = 2\pi n/L$. This is a key simplification we exploit in the following. The corresponding wave function is

$$\Phi_{R,n}(x,y) \simeq A_n e^{ip_n\frac{x+y}{2}}\left[e^{(i\kappa_n - \eta_n)|x-y|}\right.$$
$$\left. + (-1)^n e^{(i\kappa_n - \eta_n)(L-|x-y|)}\right] \ , \qquad (67)$$

where the normalization constant is

$$|A_n|^2 = \frac{1}{L\tanh(\eta_n)} + o(L^{-1}) \ . \qquad (68)$$

The wave-function (67) describes a two-particle bound state, and is the analog [21] of the $k$-$\Lambda$-string excitation in the one-dimensional Hubbard model [60, 64] in the case where the hopping amplitude is purely imaginary. The fact that we are dealing with a bound state solution implies that the associated eigenoperators (19) of the Lindbladian are sums of terms that are exponentially localized around the centre of mass between the creation and annihilation operator.

### 2. Eigenvalue spectrum for Model II

The same two classes of solutions as in Model I exists here as well. In the $L \to \infty$ limit solutions with real $k_{1,2}$ give rise to the same two-parameter continuum of states as in Model I, *cf.* Fig. 1. Complex solutions to the quantization conditions (58) have the form $q = \kappa + i\eta$ with $\kappa =$

$\frac{\pi}{2}\mathrm{sgn}\left(J\cos(\frac{p}{2})\cos(p)\right)$ and $\eta = \mathrm{arccosh}\left(\left|\frac{\gamma\cos(p)}{2J\cos(p/2)}\right|\right)$ and exists provided that $\left|\frac{2J\cos(p/2)}{\gamma\cos(p)}\right| < 1$. They correspond to momentum-dependent real eigenvalues of the form

$$\lambda(p) = -2\gamma\left(1 + \cos(p)\sqrt{1 - \left(\frac{2J\cos(p/2)}{\gamma\cos(p)}\right)^2}\right). \quad (69)$$

The dispersion (69) is shown in Fig. 3 for $\gamma = 3$ and $L = 100$. The wave function corresponding to this state in the large-$L$ limit can again be written in the form (67).

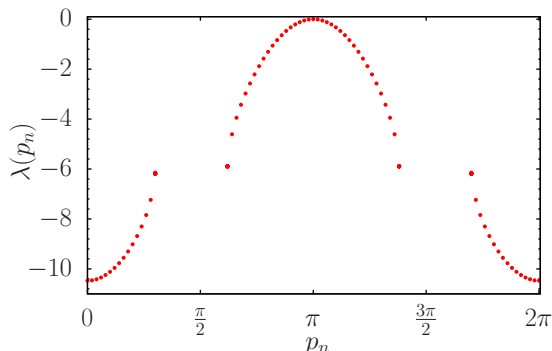

FIG. 3. Real eigenvalues corresponding to bound modes for Model II with $\gamma = 3$, $J = 1$ and $L = 100$. The eigenvalues vanish as $(p_n - \pi)^2$ for $p_n \to \pi$, reflecting the existence of a diffusive hydrodynamic mode.

### 3. Eigenvalue spectrum for Model III

In Model III we again have a two-parameter continuum of complex eigenstates in the thermodynamic limit, *cf.* Fig. 1. In contrast to Models I and II there are now two different classes of solutions with complex wavenumbers, which arise respectively from the two quantization conditions (59) and (60). The bound state solutions to eqn (59) are of the form $q = \kappa + i\eta$, where for $L \to \infty$ we find that $\kappa = -\frac{\pi}{2}\mathrm{sgn}(J\nu\cos(\frac{p}{2}))$ and $\eta = -\log\left(\left|\frac{2J\cos(p/2)}{\nu\gamma}\right|\right)$. This corresponds to $q = -i\log\left(-\frac{2iJ\cos(p/2)}{\nu\gamma}\right)$ and gives rise to real eigenvalues

$$\lambda(p) = -(2-\nu)\gamma - \frac{4J^2\cos(p/2)^2}{\nu\gamma}. \quad (70)$$

These solutions exist only if $\left|\frac{2J\cos(p/2)}{\gamma}\right| < 1$. The eigenvalues (70) are shown as functions of the total momentum in Fig. 4. For large values of $L$ the corresponding wavefunction is given by

$$\Phi_{R,n}(x,y) \simeq A_n e^{ip_n\frac{x+y}{2}}(-i)^{|x-y|}\left[\left(\frac{2J\cos(p_n/2)}{\nu\gamma}\right)^{|x-y|} - (-1)^{n+L/2+|x-y|}\left(\frac{2J\cos(p_n/2)}{\nu\gamma}\right)^{L-|x-y|}\right]. \quad (71)$$

The finite-size corrections are again exponentially small in $L$.

The second class of solutions arises from the quantization conditions (60). These can be written as a third order polynomial equation for $x = e^{i\kappa - \eta}$ (with $\eta > 0$)

$$J\gamma\nu\cos\left(\frac{p}{2}\right)x^3 - i\left(2J^2\cos\left(\frac{p}{2}\right)^2 + \nu\gamma^2\right)x^2 + J\gamma(2+\nu)\cos\left(\frac{p}{2}\right)x + 2iJ^2\cos\left(\frac{p}{2}\right)^2 = 0. \quad (72)$$

In contrast to the previous two models the eigenvalues generally have non-zero imaginary parts

$$\lambda(p) = -2\gamma - 2iJ\cos(p/2)\left(x + x^{-1}\right). \quad (73)$$

The corresponding wave function can again be expressed in the form (67). The real and imaginary parts of the eigenvalues (73) are shown as functions of the total momentum in Fig. 4.

### 4. Eigenvalue spectrum for Model IV

Model IV has the same structure as Model III. The first class of solution fulfil the quantization conditions (61). The wavenumber is $q = -i\log\left(-\frac{2iJ\cos(p/2)}{\xi\gamma}\right)$ with

real eigenvalues

$$\lambda(p) = -(2-\xi)\gamma - \frac{4J^2\cos(p/2)^2}{\xi\gamma}. \quad (74)$$

These solutions exist as long as $\left|\frac{2J\cos(p/2)}{\gamma}\right| < 1$. The corresponding wavefunction is

$$\Phi_{R,n}(x,y) = A_n e^{ip_n\frac{x+y}{2}}(-i)^{|x-y|}\left[\left(\frac{2J\cos(p_n/2)}{\xi\gamma}\right)^{|x-y|} - (-1)^{n+L/2+|x-y|}\left(\frac{2J\cos(p_n/2)}{\xi\gamma}\right)^{L-|x-y|}\right]. \quad (75)$$

The eigenvalues (74) are shown as green lines in Figs 6 and 7 for Models IV (a) and Models IV (b) respectively. We observe that the real part is always strictly negative.

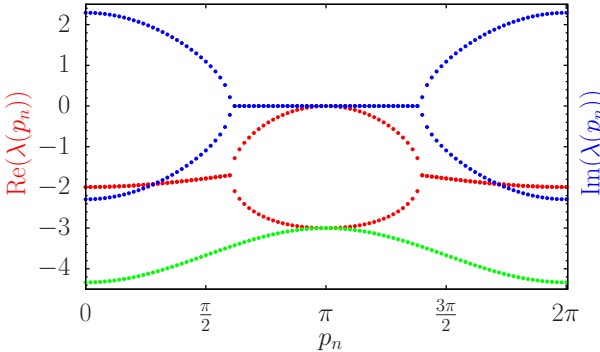

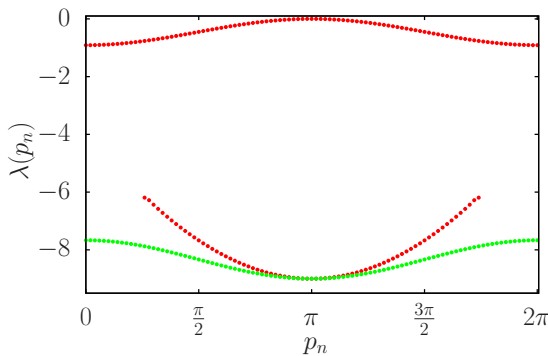

FIG. 4. Real (red) and imaginary (blue) parts of the eigenvalues (73) as a function of total momentum for Model III with $\nu = 1$, $\gamma = 3$, $J = 1$ and $L = 100$. The eigenvalues vanish as $(p_n - \pi)^2$ for $p_n \to \pi$, reflecting the existence of a diffusive hydrodynamic mode. The green symbols show the purely real eigenvalues (70) for the same parameters.

FIG. 5. Same as Fig. 4 with $\nu = -1$. Here all eigenvalues (73) are real.

The second class of solutions arise from the quantization conditions (62). They lead to a third order polynomial

$$J\xi\gamma \cos\left(\frac{p}{2}\right) x^3 + i\left(2J^2 \cos\left(\frac{p}{2}\right)^2 + \xi\gamma^2 \cos(p)\right) x^2 + J\gamma \cos\left(\frac{p}{2}\right)(2\cos(p) + \xi) x - 2iJ^2 \cos\left(\frac{p}{2}\right)^2 = 0 , \qquad (76)$$

where $x = e^{i\kappa - \eta}$ (with $\eta > 0$). Just as in Model III, the eigenvalues can have non-zero imaginary parts. They are given by

$$\lambda(p) = -2\gamma - 2iJ\cos(p/2)\left(x + x^{-1}\right) . \qquad (77)$$

The wavefunction has the same form as in Model I, II and III (67). The real and imaginary parts of the eigenvalues (77) are shown in Figs 6 and 7 for Models IV (a) and Models IV (b) respectively. We observe that the eigenvalues vanish as $(p - \pi)^2$ for $p \to \pi$, reflecting the existence of a diffusive hydrodynamic mode.

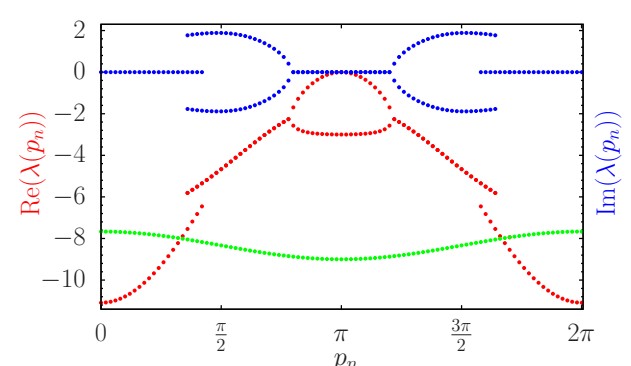

FIG. 7. Same as Fig. 6 for Model IV (b).

### B. Position representation for the eigenvalue equation with $N_\uparrow = 2$

We now turn to the time evolution of operators quadratic in fermions that carry non-zero U(1) charge

$$\hat{\Psi}_{2,0} = \sum_{x>y} \Psi(x,y) c_x^\dagger c_y^\dagger . \qquad (78)$$

The vectorized form of their equations of motion is given by (27) with $N_\downarrow = 0$, $N_\uparrow = 2$. To proceed we again solve the corresponding eigenvalue equation (33) in the position representation, which reads

$$\sum_{x',y'} \mathfrak{H}(x,y|x',y') \Phi_{R,\alpha}(x',y') = \lambda_\alpha \Phi_{R,\alpha}(x,y) . \qquad (79)$$

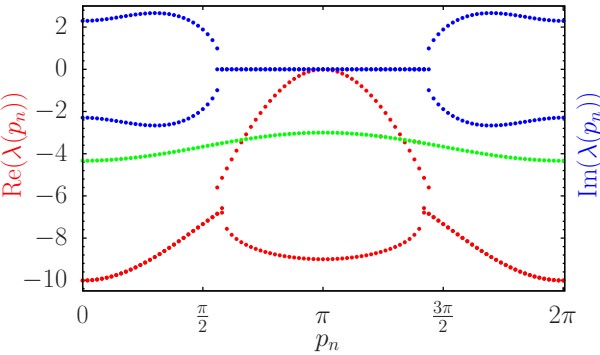

FIG. 6. Eigenvalues (74) (green) and (77) (red/blue) as functions of total momentum for Model IV (a) with $\gamma = 3$, $J = 1$ and $L = 100$. A diffusive eigenmode with quadratic dispersion is visible at $p_n \approx \pi$.

The Hamiltonian in the position representation for Model I is now given by

$$\mathfrak{H}(x,y|x',y') = -iJ\sum_{\tau=\pm 1}\left(\delta_{x,x'+\tau}\delta_{y,y'} + \delta_{x,x'}\delta_{y,y'+\tau}\right)$$
$$- 2\gamma\delta_{x,x'}\delta_{y,y'} \ , \tag{80}$$

while for Models II, III and IV we have

$$\mathfrak{H}(x,y|x',y') = -iJ\sum_{\tau=\pm 1}\left(\delta_{x,x'+\tau}\delta_{y,y'} + \delta_{x,x'}\delta_{y,y'+\tau}\right)$$
$$- \gamma\left(2 + \beta\sum_{\tau=\pm 1}\delta_{x,y+\tau}\right)\delta_{x,x'}\delta_{y,y'} \ . \tag{81}$$

The parameter $\beta = -1$ for Models II and IV, while $\beta = \nu$ for Model III. The Schrödinger equations are easily solved using a plane wave ansatz

$$\Phi_{R,\alpha}(x,y) = \begin{cases} Ae^{i(k_1 x + k_2 y)} + Be^{i(k_2 x + k_1 y)} & \text{if } x > y, \\ 0 & \text{if } x = y, \\ -\Phi_{R,\alpha}(y,x) & \text{if } x < y. \end{cases} \tag{82}$$

Inspection of the eigenvalue equation for $x \gg y$ fixes the eigenvalue as a function of $k_{1,2}$

$$\lambda_\alpha = -2\gamma - 2iJ(\cos(k_1) + \cos(k_2)) \ . \tag{83}$$

The coefficients $A$, $B$ are related by the boundary conditions

$$\Phi_{R,\alpha}(L+1, y) = -\Phi_{R,\alpha}(y, 1) \ , \tag{84}$$

which give

$$Ae^{ik_1 L} = -B \ , \quad e^{i(k_1 + k_2)L} = 1 \ . \tag{85}$$

The resulting quantization conditions for $k_{1,2}$ are:

1. Model I:

$$k_{1,2} = \frac{2\pi n_{1,2}}{L} \ , \quad -\frac{L}{2} \le n_1 < n_2 < \frac{L}{2} \ . \tag{86}$$

2. Models II, III and IV:

$$k_{1,2} = \frac{p_n}{2} \pm q_n \ , \quad p_n = \frac{2\pi n}{L} \ , \quad n = 1,\ldots,L \ ,$$
$$(-1)^n e^{iq_n L} = \frac{2J\cos(\frac{p_n}{2}) + i\beta e^{iq_n}\gamma}{2J\cos(\frac{p_n}{2}) + i\beta e^{-iq_n}\gamma} \ . \tag{87}$$

*1. Eigenvalue spectrum for Model I*

Numerical results for the finite size eigenvalue spectrum are shown in Fig. 8. We see that all eigenvalues have a real part of $-2\gamma$. This implies that all operators $\mathcal{O}(x,y)$ decay exponentially in time. This is expected for charged operators.

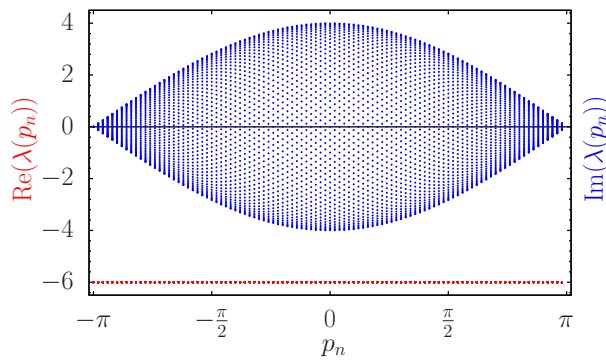

FIG. 8. Eigenvalues for $N_\uparrow = 2$ as a function of total momentum for Model I with $\gamma = 3$, $J = 1$ and $L = 100$. The real parts are $-2\gamma$ leading to strongly damping of the corresponding modes.

*2. Eigenvalue spectra for Models II, III and IV*

In Models II, III and IV, there are two classes of solutions to the quantization conditions (87). One class is the two-particle continuum discussed above for Model I and shown in Fig. 8. The second class of solutions involves complex wavenumbers, but real eigenvalues. These so-

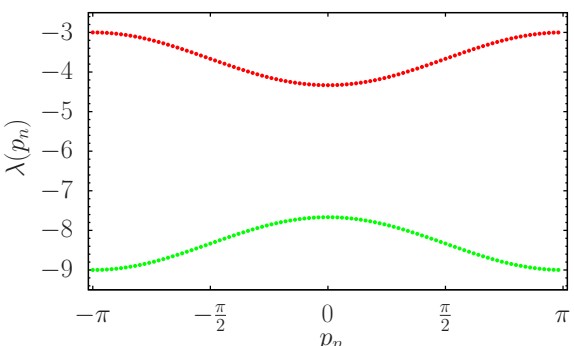

FIG. 9. Real eigenvalues for $N_\uparrow = 2$ as a function of total momentum in Model III with $\nu = +1$ (green) and Models II, III with $\nu = -1$ and IV (red) with $\gamma = 3$, $J = 1$ and $L = 100$.

lutions have the form $q = \kappa + i\eta$, where for large $L$ we have (up to exponentially small finite-size corrections) $\kappa = \frac{\pi}{2}\text{sgn}(J\beta\cos(\frac{p}{2}))$ and $\eta = -\log\left(\left|\frac{2J\cos(p/2)}{\beta\gamma}\right|\right)$. The corresponding eigenvalues are

$$\lambda(p) = -\gamma(2 + \beta) + \frac{4J^2\cos(p/2)^2}{\beta\gamma} \ . \tag{88}$$

These solutions exist as long as $\left|\frac{2J\cos(p/2)}{\gamma}\right| < 1$. For $\beta = 1$, we have that $\lambda(p) < -2\gamma$. The eigenvalues are shown for $\gamma = 3$, $J = 1$ and $L = 100$ in Fig 9. Putting everything together we obtain the following result for the

decay rate of the operators $\hat{\Psi}_{2,0}$

$$\Delta_{2,0} = \begin{cases} -\gamma & \beta = -1 \\ -2\gamma & \beta = +1 \end{cases} . \qquad (89)$$

## VI. HYDRODYNAMIC PROJECTIONS

We now turn to the problem of hydrodynamic projections in the $N_\uparrow = N_\downarrow = 1$ sector. We have seen that in the models with U(1) symmetry there exists a single-particle mode in this sector, which has a vanishing real part at $p = \pi$. Clearly this mode will dominate the late-time behaviour. The corresponding eigenstates are

$$|\Phi_{R,n}\rangle = \sum_{x,y} \Phi_{R,n}(x,y) c_{x,\uparrow}^\dagger c_{y,\downarrow}^\dagger |0\rangle , \qquad (90)$$

where the wave functions are given by (67). At late times we may therefore simplify the expression (34) for the time evolved state by restricting the sum over eigenstates to the subset of states (90)

$$|\Psi_{1,1}(t \gg \gamma^{-1})\rangle \approx \sum_n \langle \Phi_{L,n}|\Psi_{1,1}(0)\rangle e^{\lambda_n t} |\Phi_{R,n}\rangle . \qquad (91)$$

We now show how to explicitly evaluate (91) for Model I, Models II, III and IV can be dealt with in the same way. A related analysis was recently carried out for Model I to study non-equilibrium evolution from domain-wall initial conditions [30, 31]. We start by noting that the left eigenstates are simply related to the conjugates of the right eigenstates by

$$\langle \Phi_{L,\alpha}| = \langle \Phi_{R,\alpha}|\Big|_{J \to -J} . \qquad (92)$$

To simplify the discussion we now restrict our analysis to two cases:

A. Operators carrying a definite momentum: As an example we consider

$$\hat{\Psi}_1^q = \sum_x c_x^\dagger c_{x+d} e^{iqx} \qquad (93)$$

where we take $|d| \ll L$ and $q = 2\pi m/L$. These operators carry definite momentum $q$. The wave function of the corresponding state is

$$\psi_1^q(x,y) = \delta_{y,x+d} e^{iqx} . \qquad (94)$$

The overlap of the corresponding state with left eigenstates corresponding $\langle \Phi_{L,n}|$ is

$$\frac{\langle \Phi_{L,n}|\Psi_1^q(0)\rangle}{A_n^* L} = \frac{1}{A_n^* L} \sum_x \Phi_{L,n}(x, x+d) e^{iqx} (-1)^{x+d}$$

$$= (-1)^d e^{-ip_n \frac{d}{2}} \delta_{p_n, q+\pi} \begin{cases} e^{(i\kappa_n - \eta_n)|d|} & q \neq 0, \\ \delta_{d,0} & q = 0 . \end{cases} \qquad (95)$$

This leads to exponential decay of all $\Psi_1^{q \neq 0}$ with decay rate

$$\Delta(q) = -2\gamma \left[ 1 - \sqrt{1 - \left(\frac{2J}{\gamma}\sin\left(\frac{q}{2}\right)\right)^2} \right] . \qquad (96)$$

Finally, $\hat{\Psi}_1^{q=0}(t)$ has a vanishing projection on the hydrodynamic mode unless $d = 0$, in which case it is simply the (conserved) particle number. The time-evolved operators are of the form

$$\hat{\Psi}_1^q(t) = \sum_{x,y} \psi_1^q(x,y;t) c_x^\dagger c_y . \qquad (97)$$

A useful measure of the exponential decay in time is then

$$F(q,d;t) \equiv \frac{1}{L} \sum_{x,y} |\psi_1^q(x,y;t)| . \qquad (98)$$

In Fig. 10 we show the time-dependence of $F(q,d;t)$ for several values of the momentum $q$ and the separation $d$ on a log-linear scale for $L = 100$, $J = 1$ and $\gamma = 1.5$. We observe excellent agreement with the decay rate (96). For $(q,d) = (\frac{\pi}{5}, 10)$ we observe two regimes: at early

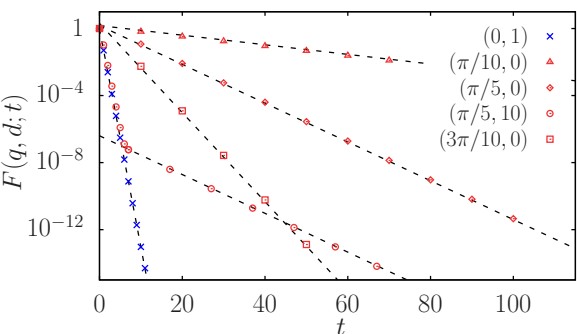

FIG. 10. Exponential decay of the operators $\hat{\Psi}_1^q(t)$ for several values of $(q, d)$, with parameters $L = 100$, $J = 1$ and $\gamma = 1.5$. The dashed lines show the predicted exponential decay rates (96).

times the operator decays exponentially with a rate of approximately $-2\gamma$, which is set by the real part of the two-particle continuum shown in Fig. 1. At late times the decay rate is set by the bound state solution shown in Fig. 2. As we are considering an operator with definite momentum, only a single momentum of the bound state solution will contribute (as long as the latter exists). However, this has a very small overlap with the operator under consideration, which is why this behaviour only becomes visible at late times, when the contributions due to the two-particle continuum have already decayed to negligible values.

Our results show that quadratic operators carrying a definite momentum do not acquire diffusive power-law (in $t$) tails. This is a result of the kinematics of the

hydrodynamic modes, which is a direct consequence of the decoupling of the BBGKY hierarchy.

B. Spatially local operators: We focus on the operators $\hat{\Psi}_{x_0,y_0} = c_{x_0}^\dagger c_{y_0}$, where we restrict the positions to fulfil $|x_0 - y_0| \ll L$ and $1 \ll x_0, y_0 \ll L$. The wave function $\psi_{x_0,y_0}(x,y)$ of the corresponding state $|\Psi_{x_0,y_0}(0)\rangle$ is

$$\psi_{x_0,y_0}(x,y) = \delta_{x,x_0}\delta_{y,y_0} \ . \tag{99}$$

Dropping terms that are exponentially small in $L$ we find

$$\langle \Phi_{L,n} | \Psi_{x_0,y_0}(0) \rangle = \Phi_{L,n}(x_0,y_0)(-1)^{y_0}$$
$$\simeq (-1)^{y_0} A_n^* e^{-ip_n \frac{x_0+y_0}{2}} e^{(i\kappa_n - \eta_n)|x_0-y_0|}. \tag{100}$$

For late times $\gamma t \gg 1$ we then have

$$\psi_{x_0,y_0}(x,y;t) \simeq \sum_n \langle \Phi_{L,n}|\Psi_{x_0,y_0}(0)\rangle e^{\lambda_n t}\Phi_{R,n}(x,y)(-1)^y$$
$$\simeq \frac{(-1)^{y+y_0}}{L}\sum_n \coth\eta_n e^{\lambda_n t + i\frac{p_n}{2}Y + (i\kappa_n-\eta_n)X}, \tag{101}$$

where we have defined

$$X = |x-y| + |x_0 - y_0| \ , \quad Y = (x-x_0) + (y-y_0) \ . \tag{102}$$

For large $L$ we can turn the sum into an integral

$$\psi_{x_0,y_0}(x,y;t) \simeq (-1)^{y+y_0}\int_{p_0}^{2\pi-p_0}\frac{\mathrm{d}p}{2\pi}\coth(\eta(p))e^{\lambda(p)t+i\frac{p}{2}Y}$$
$$\times \ e^{(i\kappa(p)-\eta(p))X} \ , \tag{103}$$

where we have assumed that $J > 0$ and defined

$$\lambda(p) = -2\gamma\left[1 - \sqrt{1 - \left(\frac{2J}{\gamma}\cos\left(\frac{p}{2}\right)\right)^2}\right] \ ,$$

$$\eta(p) = \mathrm{arccosh}\left(\left|\frac{\gamma}{2J\cos(p/2)}\right|\right) \ ,$$

$$\kappa(p) = -\frac{\pi}{2}\mathrm{sgn}\left(\cos\left(\frac{p}{2}\right)\right) \ ,$$

$$p_0 = 2\arccos(\frac{\gamma}{2J})\ \theta(2J - \gamma) \tag{104}$$

In order to extract the late time asymptotics of the integral it is useful to introduce

$$z(p) = \frac{2J\cos(p/2)}{\gamma} \ , \tag{105}$$

and note that for $0 < z \ll 1$

$$e^{-\eta(p)} = \frac{z}{2}\left(1 + \frac{z^2}{4} + \frac{z^4}{8} + \dots\right) \ ,$$

$$\coth(\eta(p)) = 1 + \frac{z^2}{2} + \frac{3z^4}{8} + \dots \tag{106}$$

Folding the integral around $\pi$, and expanding the integrand around $p = \pi$ we obtain

$$\psi_{x_0,y_0}(x,y;t) \simeq (-1)^{y+y_0}\int_{p_0}^{\pi}\frac{\mathrm{d}p}{2\pi}\left[\frac{J(\pi-p)}{2\gamma}\right]^X e^{-Dt(\pi-p)^2}$$
$$\times\ c(\pi-p)\left[e^{i\frac{p}{2}Y-i\frac{\pi}{2}X} + (-1)^Y e^{-i\frac{p}{2}Y+i\frac{\pi}{2}X}\right] \ . \tag{107}$$

Here we have defined

$$c(q) = 1 + \left[\frac{J^2}{2\gamma^2} - \frac{X}{24}\left(1 - \frac{6J^2}{\gamma^2}\right)\right]q^2 + \dots \ ,$$

$$D = \frac{J^2}{\gamma} \ . \tag{108}$$

Finally we change the integration variable to $q = \pi - p$ and extend the upper integration boundary to infinity. The integral then can be expressed in terms of confluent hypergeometric functions

$$\psi_{x_0,y_0}(x,y;t) \approx (-1)^{\min(x-y,0)+\max(x_0-y_0,0)}\frac{1}{2\pi}\left(\frac{J}{2\gamma}\right)^X\left[g(X,Dt,Y) + \frac{1}{2}\frac{\mathrm{d}^2 c(q)}{\mathrm{d}q^2}\bigg|_{q=0}g(X+2,Dt,Y) + \dots\right],$$

$$g(n,Dt,Y) = \frac{n!}{2^n}(Dt)^{-\frac{1+n}{2}}\begin{cases} \mathrm{Re}\left(U(\frac{1+n}{2},\frac{1}{2},-\frac{Y^2}{16Dt})\right) & \text{if } Y\text{ even,} \\ i\,\mathrm{sgn}(Y)\,\mathrm{Im}\left(U(\frac{1+n}{2},\frac{1}{2},-\frac{Y^2}{16Dt})\right) & \text{if } Y\text{ odd.} \end{cases} \tag{109}$$

This exhibits power-law tails at late times (for fixed $x$, $y$)

$$\psi_{x_0,y_0}(x,y;t)\bigg|_{\text{leading}} = \begin{cases} A_e\ (Dt)^{-\frac{1+X}{2}} & \text{Y even} \ , \\ A_o\ (Dt)^{-\frac{2+X}{2}} & \text{Y odd} \ , \end{cases} \tag{110}$$

where

$$A_e = (-1)^{\min(x-y,0)+\max(x_0-y_0,0)} \frac{1}{2\pi}\left(\frac{J}{2\gamma}\right)^X \Gamma\left(\frac{1+X}{2}\right),$$

$$A_o = -iA_e Y \frac{\Gamma(1+\frac{X}{2})}{2\Gamma(\frac{1+X}{2})} . \tag{111}$$

Some of these power-laws were recently calculated in [48] using a transfer matrix approach. To summarize, we have established that at late times $\gamma t \gg 1$ the Heisenberg-picture operator takes the following form

$$\left(c_{x_0}^\dagger c_{y_0}\right)(t) \simeq \sum_{x,y} \psi_{x_0,y_0}(x,y;t) c_x^\dagger c_y , \tag{112}$$

where $\psi_{x_0,y_0}(x,y;t)$ is given by (109). This is a key result of our work. The slowest decaying operator is obtained by taking $x_0 = y_0$, which at late times behaves as

$$\left(c_{x_0}^\dagger c_{x_0}\right)(t) = \sum_{x} \psi_{x_0,x_0}(x,x;t) c_x^\dagger c_x + \dots \tag{113}$$

This operator spreads diffusively, as is shown in Fig. 11 (see also [65]).

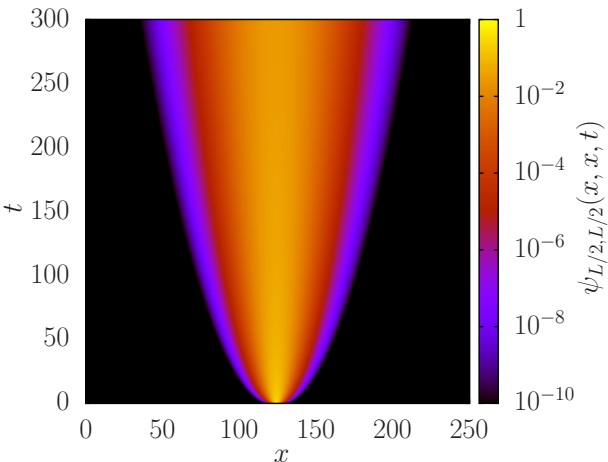

FIG. 11. Diffusive spreading of the operator $\left(c_{L/2}^\dagger c_{L/2}\right)(t)$, with $L = 250$, $J = 1$ and $\gamma = 3$.

In Figs 12 and 13 we compare the analytic expressions (109) and (110) to numerically exact results for the amplitudes $\psi_{x_0,y_0}(x,y;t)$.

## A. Hydrodynamic projections in Models II, III, IV

Hydrodynamic projections in Models II, III and IV can be determined along the same lines as in Model I. The wave function of the hydrodynamic mode has the same form in terms of the parameters $\kappa_n$ and $\eta_n$, and all we have to do is to substitute the appropriate expressions for these in Models II-IV.

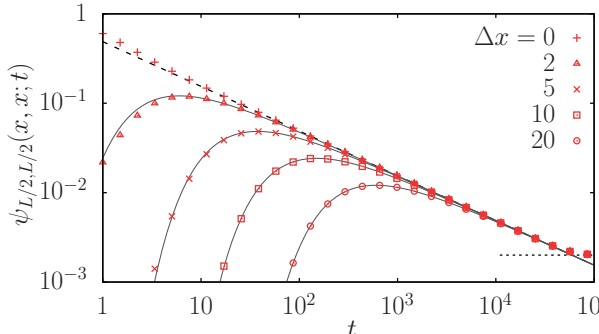

FIG. 12. Amplitude $\psi_{L/2,L/2}(x,x;t)$ with $x = \frac{L}{2} + \Delta x$ of the operator $(c_{L/2}^\dagger c_{L/2})(t)$ for $L = 500$, $J = 1$ and $\gamma = 3$. The symbols, continuous lines and dashed lines show respectively the numerically exact result, the analytic expression (109) and the leading power law (110). The stationary value $1/L$ reached at late times is shown by the dotted line.

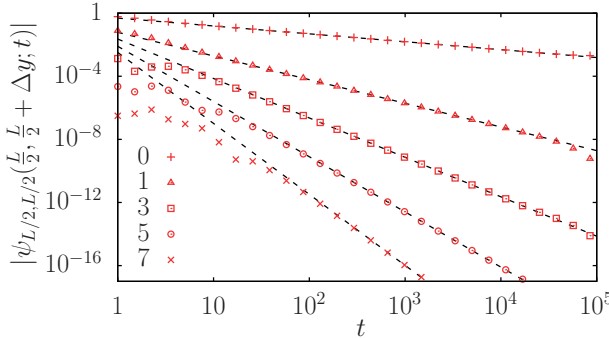

FIG. 13. Amplitude of $\left(c_{L/2}^\dagger c_{L/2}\right)(t)$, with $L = 500$, $J = 1$ and $\gamma = 3$. The plot shows the amplitude at different $\Delta y$ values. The dashed lines show the leading power law behaviour (110), $t^{-1/2}$, $t^{-3/2}$, $t^{-5/2}$, $t^{-7/2}$, $t^{-9/2}$ respectively.

A. Operators carrying a definite momentum: The operators $\hat{\Psi}_1^q$ defined in (93) decay exponentially in time. The decay rate $\Delta(q)$ is given in terms of the eigenvalues $\lambda(p)$ (69) (Model II),(70), (73) (Model III) and (74), (77) (Model IV) as $\Delta(q) = \max(\text{Re}(\lambda(q+\pi)))$. Here the maximum is taken over the different modes at fixed momentum. In Model II a simple closed-form expression is available

$$\Delta(q) = -2\gamma\left[1 - \cos(q)\sqrt{1 - \left(\frac{2J\sin(q/2)}{\gamma\cos(q)}\right)^2}\right] . \tag{114}$$

As for Model I, the projection of $\hat{\Psi}_1^0$ on the hydrodynamic mode vanishes unless $d = 0$, in which case it reduces to the conserved particle number.

B. Spatially local operators: The wave-functions describing the evolution of the operators $\hat{\Psi}_2 = c_{x_0}^\dagger c_{y_0}$ at late times are again of the form (103). The difference com-

pared to our analysis for Model I arises from the expansions of $\lambda(p)$ and $\eta(p)$ around $p = \pi$. In Model II we obtain

$$\lambda(p) = -\frac{J^2 + \gamma^2}{\gamma}(\pi - p)^2 + \dots \, ,$$

$$e^{-\eta(p)} = \frac{J(\pi - p)}{2\gamma}\Big(1 + \frac{6J^2 + 11\gamma^2}{24\gamma^2}(\pi - p)^2 + \dots\Big) \, ,$$

$$\coth \eta(p) = 1 + \frac{J^2}{2\gamma^2}(\pi - p)^2 + \dots \, . \tag{115}$$

Substituting this into the integral representation (107) leads to the same form (109) for the late time asymptotics, with the replacements

$$c(q) = 1 + \Big[\frac{J^2}{2\gamma^2} + \frac{X}{24}\Big(\frac{6J^2}{\gamma^2} + 11\Big)\Big]q^2 + \dots \, ,$$

$$D = \frac{(J^2 + \gamma^2)}{\gamma} \, . \tag{116}$$

The same analysis for Models III and IV leads again to (109), where $D$ and $c(q)$ are now given by

Model III:

$$c(q) = 1 + \Big[\frac{J^2}{2\gamma^2} - \frac{X}{24}\Big(1 - \frac{6J^2(2+\nu)}{\gamma^2(2-\nu)}\Big)\Big]q^2 + \dots \, ,$$

$$D = \frac{2J^2}{(2-\nu)\gamma} \, , \tag{117}$$

Model IV:

$$c(q) = 1 + \Big[\frac{J^2}{2\gamma^2} + \frac{X}{24}\Big(11 + \frac{6J^2(2-\xi)}{\gamma^2(2+\xi)}\Big)\Big]q^2 + \dots \, ,$$

$$D = \frac{2J^2 + (2+\xi)\gamma^2}{(2+\xi)\gamma} \, . \tag{118}$$

All models show the same leading behaviour (110), with appropriately rescaled time. Differences arise in the subleading terms. Note that taking the limit $\nu \to 0$ or $\xi \to 0$ recovers the results for Models I and II, as expected.

## VII. DYNAMICS OF OPERATORS QUADRATIC IN FERMIONS IN MODEL V

We now turn to Model V, which does not have particle number conservation and we therefore do not expect any hydrodynamic behaviour at late times. On the other hand, the absence of particle number conservation makes the solution of the equations of motion in the Heisenberg picture more interesting. The vectorized Hamiltonian (32) has the key property that its interaction terms in $\mathcal{H}$ contain at least as many annihilation operators as creation operators. This results in a triangular structure of the corresponding eigenvalue equations, which facilitates their solution.

### A. Position representation for the eigenvalue equation with at most $N_\uparrow = N_\downarrow = 1$ fermions

Our aim is to solve the time evolution equation for the operator

$$\hat{\Psi}_{1,1} = \sum_{x,y} \Psi(x,y) c_x^\dagger c_y \, . \tag{119}$$

In our vectorized notations this reads

$$|\Psi_{1,1}\rangle = \sum_{x,y} \Psi(x,y)(-1)^y c_{x,\uparrow}^\dagger c_{y,\downarrow}^\dagger |0\rangle \, . \tag{120}$$

The eigenvalue equations of the Hamiltonian $\mathcal{H}$ needed to work out $|\Psi_{1,1}(t)\rangle$ are

$$\mathcal{H}|0\rangle = 0 \, ,$$

$$\mathcal{H}|\Phi_{R,\alpha}\rangle = \lambda_\alpha |\Phi_{R,\alpha}\rangle \, , \tag{121}$$

where

$$|\Phi_{R,\alpha}\rangle = \sum_{x,y} \Phi_{R,\alpha}(x,y) c_{x,\uparrow}^\dagger c_{y,\downarrow}^\dagger |0\rangle + v_\alpha |0\rangle \, . \tag{122}$$

We then have

$$|\Psi_{1,1}(t)\rangle = \sum_\alpha \langle \Phi_{L,\alpha}|\Psi_{1,1}\rangle \, e^{\lambda_\alpha t}|\Phi_{R,\alpha}\rangle \, , \tag{123}$$

which includes a contribution proportional to the identity. The position representation of (121) reads

$$\sum_{x',y'} \mathfrak{H}(x,y|x',y')\Phi_{R,\alpha}(x',y') = \lambda_\alpha \Phi_{R,\alpha}(x,y) \, ,$$

$$2\gamma \sum_x (-1)^x \Phi_{R,\alpha}(x,x) = \lambda_\alpha v_\alpha \, , \tag{124}$$

where

$$\mathfrak{H}(x,y|x',y') = -iJ \sum_{\tau=\pm1} (\delta_{x,x'+\tau}\delta_{y,y'} + \delta_{x,x'}\delta_{y,y'+\tau})$$

$$- 2\gamma\Big(1 - \sum_{\tau=\pm1} \delta_{x,y+\tau}\Big)\delta_{x,x'}\delta_{y,y'}$$

$$+ \delta_{x,y} \sum_{\tau=\pm1} \delta_{x+\tau,x'}\delta_{y+\tau,y'} \, . \tag{125}$$

To solve this, we can use the same plane wave ansatz (49) as in the $U(1)$ symmetric models. The eigenvalue and the boundary conditions are the same as in the $U(1)$ symmetric models (50) and (51). We find two solutions, either $B = -Ae^{ik_1 L}$ and the relative momentum is quantised by

$$(-1)^n e^{iq_n L} = \frac{2J\cos(p_n/2) - ie^{iq_n}\gamma}{2J\cos(p_n/2) - ie^{-iq_n}\gamma} \, , \tag{126}$$

or $B = Ae^{ik_1 L}$ and

$$(-1)^n e^{iq_n L} = \frac{2J\cos(p_n/2)(2J\cos(p_n/2)\sin(q_n) + \gamma\cos(p_n)) + e^{iq_n}\gamma(2J\cos(p_n/2)\cos(q_n) - i\gamma\cos(p_n))}{2J\cos(p_n/2)(2J\cos(p_n/2)\sin(q_n) - \gamma\cos(p_n)) - e^{-iq_n}\gamma(2J\cos(p_n/2)\cos(q_n) - i\gamma\cos(p_n))} . \quad (127)$$

We note that (127) is the same as in Model IV (a) with the replacement $\gamma \to -\gamma$. It turns out that there is only a single eigenstate (122) for which $v_\alpha \neq 0$, namely

$$\Phi_{R,0}(x,y) = A_0(-1)^y \delta_{x,y} , \quad v_0 = -A_0\frac{L}{2}. \quad (128)$$

The corresponding eigenoperator of the dual Lindbladian has eigenvalue $\lambda_0 = -4\gamma$ and is given by

$$\hat{\Phi}_0 = A_0\Big(\sum_x c_x^\dagger c_x - \frac{L}{2}\mathbb{1}\Big) . \quad (129)$$

To see that $v_\alpha = 0$ for all other eigenoperators we use that the Lindbladian time evolution is trace preserving

$$\mathrm{Tr}(\hat{\Phi}_\alpha(t)) = e^{\lambda_\alpha t}\mathrm{Tr}(\hat{\Phi}_\alpha) . \quad (130)$$

All eigenvalues are non-zero, which implies that all eigenoperators must be traceless. This implies that

$$v_\alpha = -\frac{1}{2}\sum_x (-1)^x \Phi_{R,\alpha}(x,x) . \quad (131)$$

Substituting (131) into the eigenvalue equation (124) imposes that either $v_\alpha = 0$ or $\lambda_\alpha = -4\gamma$. We find numerically that the eigenvalue $-4\gamma$ is non-degenerate. The corresponding eigenoperator is given above, which establishes our assertion.

### 1. Eigenvalue spectrum

The eigenvalue spectrum is easily computed numerically. Like in Models I-IV, there is a continuum of states with real wavenumbers *cf.* Fig. 1. In addition there are two classes of eigenvalues with complex wavenumbers, which are shown in Fig. 14. The first class of solutions arises from the quantization condition (126) with wavenumbers $q = -i\log\Big(-\frac{2iJ\cos(p/2)}{\gamma}\Big)$. The corresponding eigenvalues are real, and up to finite-size corrections take the simple form

$$\lambda(p) = -\gamma - \frac{4J^2\cos(p/2)^2}{\gamma} . \quad (132)$$

These solutions exist as long as $|2J\cos(p/2)/\gamma| < 1$, and the corresponding wavefunction is, up to finite-size corrections, given by

$$\Phi_{R,n}(x,y) = A_n e^{ip_n\frac{x+y}{2}}(-i)^{|x-y|}\left[\left(\frac{2J\cos(p_n/2)}{\gamma}\right)^{|x-y|}\right.$$
$$\left. -(-1)^{n+L/2+|x-y|}\left(\frac{2J\cos(p_n/2)}{\gamma}\right)^{L-|x-y|}\right] . \quad (133)$$

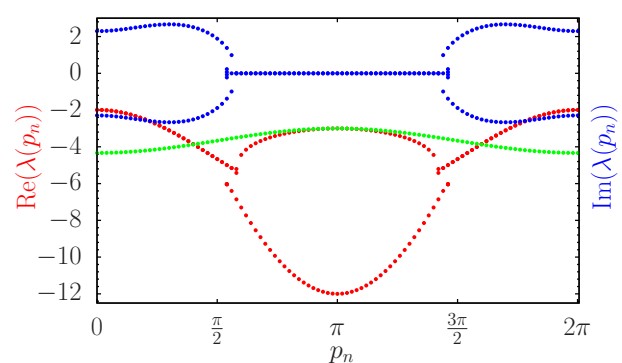

FIG. 14. Real (red) and imaginary (blue) parts of the eigenvalues (127) as a function of total momentum for Model V with $\nu = 1$, $\gamma = 3$, $J = 1$ and $L = 100$. The green symbols show the purely real eigenvalues (126) for the same parameters.

The second class of solutions arises from the quantization conditions (127). These can be cast in the form of a third order polynomial equation for $x = e^{i\kappa - \eta}$ with $\eta > 0$

$$J\gamma\cos\left(\frac{p}{2}\right)x^3 - i\Big(2J^2\cos\left(\frac{p}{2}\right)^2 + \gamma^2\cos(p)\Big)x^2$$
$$+ J\gamma\cos\left(\frac{p}{2}\right)(2\cos(p) + 1)x + 2iJ^2\cos\left(\frac{p}{2}\right)^2 = 0 . \quad (134)$$

The eigenvalues are (up to finite size corrections)

$$\lambda(p) = -2\gamma - 2iJ\cos(p/2)\left(x + x^{-1}\right) , \quad (135)$$

and just as in Models III and IV $\lambda(p)$ can have a non-zero imaginary part. The corresponding wave functions have the same form as in Models I-IV (67). As all eigenvalues have strictly negative real parts, all operators decay exponentially.

A physical process that could be analyzed in model V would be the time evolution of particle density and its fluctuations in a quantum quench starting from an initially empty state.

## VIII. OPERATORS CUBIC IN FERMIONS

Operators cubic in fermions cannot have hydrodynamic projections as a result of their odd fermion parity and are therefore expected to decay exponentially in time. We start by considering the following class of operators

$$\hat{\Psi}_{0,3} = \sum_{j<k<l} \Psi(j,k,l)c_j c_k c_l . \quad (136)$$

In Model I, the corresponding equation of motion maps onto an imaginary time Schrödinger equation for three non-interacting particles and it is easy to see that the decay rate for $\hat{\Psi}_{0,3}$ is

$$\Delta_{0,3} = -3\gamma \ . \tag{137}$$

In Models II-IV we have determined the spectrum numerically and observe that

$$\Delta_{0,3} = \begin{cases} -\gamma - \frac{2J^2}{\gamma} & \beta = -1 \\ -3\gamma & \beta = +1 \end{cases} , \tag{138}$$

where we recall that $\beta = -1$ for Models II and IV, while $\beta = \nu = \pm 1$ for Model III.

The second class of operators we consider are of the form

$$\hat{\Psi}_{1,2} = \sum_{j;k<l} \Psi(j,k,l) c_j^\dagger c_k c_l \ . \tag{139}$$

We find the following result (analytically for Model I and numerically for Models II-IV) for the decay rate of these operators

$$\Delta_{1,2} = -\gamma. \tag{140}$$

## IX.   OPERATORS QUARTIC IN FERMIONS

Finally we consider operators quartic in fermions. Arguably the most interesting operators are $U(1)$ symmetric ones

$$\hat{\Psi}_{2,2} = \sum_{j<k,l<n} \psi(j,k,l,n) c_j^\dagger c_k^\dagger c_l c_n \ . \tag{141}$$

Under time evolution these are expected to acquire hydrodynamic power-law tails at late times. In terms of the vectorized notations (26) the equations of motion for $\hat{\Psi}_{2,2}$ map onto a four-particle imaginary time Schrödinger equation. For Model I this can in principle still be solved exactly by using integrability methods [60]. However, in Models II, III and IV exact results can no longer be obtained following the same steps as in the two-particle sector, because the wave-functions of eigenstates of the Hamiltonians $\mathcal{H}$ no longer take simple forms in terms of superpositions of plane waves. This is because the two existing conserved quantities – momentum and the eigenvalue of $\mathcal{H}$ – are no longer sufficient to fix a set of four single-particle rapidities. As a result, in contrast to the situation in the two-particle sector, the wave functions of four-particle eigenstates in these models no longer takes (nested) Bethe ansatz form. We expect this to lead to quantifiably different behaviours of appropriate quantities related to $\hat{\Psi}_{2,2}(t)$ in Models II-IV compared to Model I. However, this line of enquiry goes beyond the scope of the present work and will be pursued elsewhere.

## A.   Model I: Bethe ansatz

As was noted in [21] the general eigenstates of $\mathcal{H}$ (28) in Model I have Bethe ansatz form and are parametrized in terms on $N = N_\uparrow + N_\downarrow$ rapidity variables $\{k_j | j = 1, \ldots, N\}$ and $M = N_\downarrow$ rapidity variables $\{\Lambda_\alpha | \alpha = 1, \ldots, M\}$. The two sets of rapidities fulfil the nested Bethe ansatz equations

$$e^{ik_j L} = \prod_{\alpha=1}^M \frac{\Lambda_\alpha - \sin k_j + \gamma/2}{\Lambda_\alpha - \sin k_j - \gamma/2} \ ,$$

$$\prod_{j=1}^N \frac{\Lambda_\alpha - \sin k_j + \gamma/2}{\Lambda_\alpha - \sin k_j - \gamma/2} = \prod_{\beta \neq \alpha} \frac{\Lambda_\beta - \Lambda_\alpha + \gamma}{\Lambda_\beta - \Lambda_\alpha - \gamma} \ . \tag{142}$$

The eigenvalue of $\mathcal{H}$ for a given solution is

$$\lambda(\{k_j; \Lambda_\alpha\}) = -2iJ \sum_{j=1}^N \cos k_j - N\gamma. \tag{143}$$

In the case $N_\uparrow = N_\downarrow = 1$ (142) and (143) recover (56) and (50) as required. The solutions of (142) corresponding to the hydrodynamic modes were identified in Ref [21] and are given in terms of the dissipative analog of $k$-$\Lambda$ strings [60, 64], which up to finite-size corrections take the form

$$k_{\alpha,j}^{(m)} = \arcsin(i\lambda_\alpha^{(m)} - (m - 2j + 2)\frac{\gamma}{2}),$$

$$k_{\alpha,j+m}^{(m)} = \pi - \arcsin(i\lambda_\alpha^{(m)} + (m - 2j + 2)\frac{\gamma}{2}),$$

$$\Lambda_{\alpha,j}^{(m)} = i\lambda_\alpha^{(m)} + \frac{\gamma}{2}(m + 1 - 2j), \ 1 \leq j \leq m. \tag{144}$$

The eigenvalues of $\mathcal{H}$ are expressed in terms of the string centres as

$$\lambda = 4 \sum_{(m,\alpha)} \text{Im}\sqrt{1 - \left(i|\lambda_\alpha^{(m)}| - m\frac{\gamma}{2}\right)^2} - \gamma N. \tag{145}$$

In the 4-particle sector we need to consider four classes of solutions:

1. A single $k$-$\Lambda$ string of length 2

   This corresponds to taking $m = 2$ in (144), (145). Up to finite-size corrections the corresponding eigenvalues can be expressed in terms of the total momentum $p$

   $$\lambda(p) = -4\gamma\left[1 - \sqrt{1 - \left(\frac{\sin(p/2)}{\gamma}\right)^2}\right] \ . \tag{146}$$

   This goes to zero as $p^2$ for small momenta, signalling the diffusive nature of this mode.

2. Two $k$-$\Lambda$ strings of length 1

   This corresponds to combining two strings with $m = 1$ in (144), (145). The string centres $\lambda_{1,2}^{(1)}$ fulfil

the Bethe equations [21]

$$Lf_1(\lambda_\alpha^{(1)}) = 2\pi J_\alpha^{(1)} + \sum_{\beta \neq \alpha} \theta\left(\frac{\lambda_\alpha^{(1)} - \lambda_\beta^{(1)}}{\gamma}\right), \qquad (147)$$

where $\theta(x) = 2\arctan(x)$ and $f_1(x) = \text{sgn}(x)\big(\pi - \arcsin(ix+\gamma/2)+\arcsin(ix-\gamma/2)\big)$. The (half-odd) integers $J_\alpha^{(1)}$ have range

$$|J_\alpha^{(1)}| \leq \frac{L-3}{2}, \qquad (148)$$

and the total momentum is $p = \frac{2\pi}{L}(J_1^{(1)} + J_2^{(1)})$.

3. $\eta$-pairing descendants [61, 62] of a length-1 $k$-$\Lambda$ string

   In terms of our vectorized notations these eigenstates take the form

$$\eta^\dagger|\lambda^{(1)}\rangle, \quad \eta^\dagger = \sum_j (-1)^j c_{j,\downarrow}^\dagger c_{j,\uparrow}^\dagger, \qquad (149)$$

   where $|\lambda^{(1)}\rangle$ denotes eigenstates in the $N_\uparrow = N_\downarrow = 1$ sector corresponding to length-1 $k$-$\Lambda$ string solutions to the Bethe equations (142), i.e.

$$Lf_1(\lambda^{(1)}) = 2\pi J^{(1)}, \quad |J^{(1)}| \leq \frac{L}{2} - 1. \qquad (150)$$

   The eigenvalues of these states can be expressed (up to finite-size corrections) in terms of the total momentum $p$ as

$$\lambda(p) = -2\gamma\left[1 - \sqrt{1 - \left(\frac{2\sin(p/2)}{\gamma}\right)^2}\right]. \qquad (151)$$

4. $\eta$-pairing descendant of the vacuum state

$$(\eta^\dagger)^2|0\rangle \qquad (152)$$

   This state has eigenvalue zero.

In Fig. 15 we show exact diagonalization results for the eigenvalues of $\mathcal{H}$ in the 4-particle sector with $N_\uparrow = N_\downarrow = 2$ in Model I, for parameter values $\gamma = 3$, $J = 1$ and $L = 26$. We observe that the eigenvalues with small real parts are given in terms of the four classes of Bethe states discussed above in the framework on the string hypothesis. In addition there are eigenstates with larger negative real parts, which are given in terms of other solutions of the Bethe equations (142).

In Fig. 16 we compare the eigenvalues obtained by using the string hypothesis as outlined above for $\gamma = 2.5$, $J = 1$ and $L = 26$ to results obtained by exact diagonalization of the Hamiltonian $\mathcal{H}$ in the 4-particle sector. We observe excellent agreement for all eigenvalues. This shows that it is possible to extend our analysis of section VI to the four-particle sector and determine exact hydrodynamic projections of U(1) invariant operators quartic in fermions for Model I. However, this is beyond the scope of our present work.

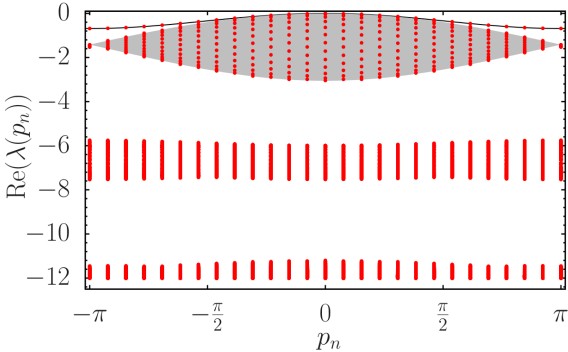

FIG. 15. Real parts of the eigenvalues of $\mathcal{H}$ with $N_\uparrow = N_\downarrow = 2$ as a function of total momentum for Model I with $\gamma = 3$, $J = 1$ and $L = 26$. The black line is the eigenvalue of a length-2 $k-\Lambda$ string (class 1 in main text), and the gray area shows the continuum of two length-1 $k-\Lambda$ strings (class 2 in main text).

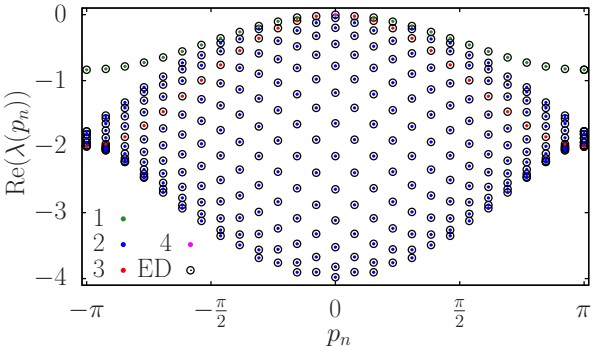

FIG. 16. Comparison of results obtained by the string hypothesis to exact diagonalization (circles) for the real parts of the eigenvalues as a function of total momentum for Model I with $\gamma = 2.5$, $J = 1$ and $L = 26$. The four different classes of solutions (see main text) are shown as green, blue, red, and pink dots.

### B. Non-integrable models

In Models II-IV we still expect hydrodynamic modes, but as these models are not integrable a detailed analytic understanding of the eigenvalues of the various Hamiltonians $\mathcal{H}$ is not available. On the other hand, the numerical spectrum of eigenvalues can be straightforwardly obtained by exact diagonalization techniques. As an example we show results for Model II with $\gamma = 3$, $J = 1$ and $L = 26$ in Fig. 17. The eigenvalues with the largest real parts form a 2-particle scattering continuum, which we observe to approximately coincide with the continuum obtained by treating the eigenvalues in the $N_\uparrow = N_\downarrow = 1$ sector (69) as corresponding to non-interacting "excitations" and adding two of them in a restricted range of momenta $p$, which we fix by requiring that $\lambda(p) \geq -\gamma$. This requirement is motivated by the observation that

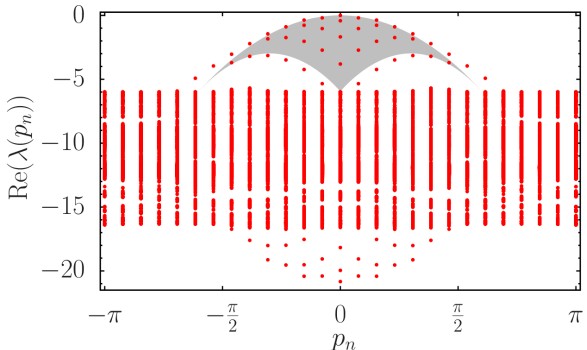

FIG. 17. Real parts of the eigenvalues of $\mathcal{H}$ in the sector $N_\uparrow = N_\downarrow = 2$ as a function of total momentum for Model II with $\gamma = 3$, $J = 1$ and $L = 26$. The gray area shows the continuum of two non-interacting $N_\uparrow = N_\downarrow = 1$ "excitations".

other eigenvalues from a continuum of states starting at $-2\gamma$, *cf.* Fig. 17. It would be interesting to pursue a more precise description by treating the "particles" as interacting and extracting their "scattering length" from the finite-size spectrum of the Lindbladian. The analogous analysis for the two-particle scattering continuum on the non-integrable spin-1 chain was carried out in Ref. [66].

### C.  Non $U(1)$ symmetric operators

Operators that are not U(1) symmetric are expected to decay exponentially in time. We first consider operators involving four annihilation operators. In Model I the corresponding imaginary-time Schrödinger equation describes non-interacting particles, and all eigenvalues have $\text{Re}(\lambda_\alpha) = -4\gamma$. In Models II-IV we have

$$\Delta_{0,4} = \begin{cases} -\gamma & \beta = -1 \\ -4\gamma & \beta = +1 \end{cases} . \tag{153}$$

For operators involving a single creation and three annihilation operators, we can only investigate the spectrum numerically and find that

$$\Delta_{1,3} = \begin{cases} -2\gamma & \text{Models I, III } (\nu = +1) \\ -\gamma & \text{Models II, IV, III } (\nu = -1) \end{cases} . \tag{154}$$

### X.  NON-EQUAL TIME CORRELATION FUNCTIONS IN THE STEADY STATE

Using the results obtained above we can evaluate non-equal time correlation functions of quadratic operators in the infinite temperature steady state, e.g.

$$g(x, y | x_0, y_0; t) = \frac{1}{2^L} \text{Tr}\left[\left(c_{x_0}^\dagger c_{y_0}\right)(t) c_x^\dagger c_y\right] - \frac{1}{4}\delta_{x,y}\delta_{x_0,y_0} . \tag{155}$$

This has a simple form in terms of wave-functions

$$g(x, y | x_0, y_0; t) = \frac{1}{4}\psi_{x_0,y_0}(y, x; t) . \tag{156}$$

As a particular example we consider the Fourier transform of the connected density-density correlation function

$$G(q, \omega) = \sum_x \int_0^\infty dt \; e^{i(\omega t - qx)} g(x, x | 0, 0; t) . \tag{157}$$

As a result of the presence of a diffusive zero energy mode in Models I-IV, $G(q, \omega)$ diverges at small momentum and energy like

$$G(q, \omega) \propto \frac{1}{i\omega + Dq^2} . \tag{158}$$

Here the model dependent parameter $D$ was previously obtained in our derivation of hydrodynamic projections in (108), (116), (117) and (118) respectively. In Figs 18 and 19 we show numerical results for $\text{Im}(G(q, \omega))$ in respectively Model I and II, for system size $L = 500$ and parameters $J = 1$ and $\gamma = 3$. The diffusive mode is

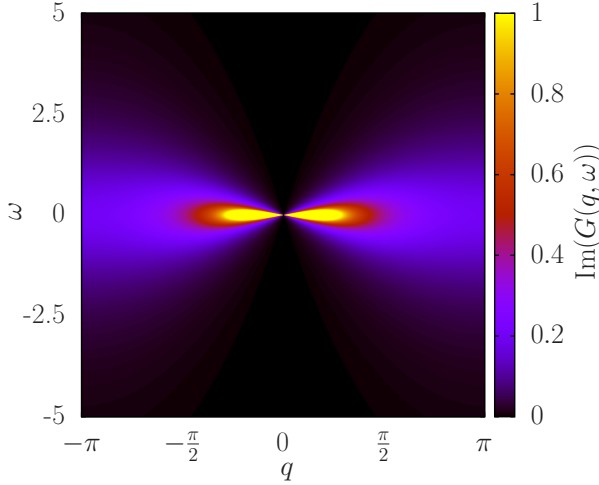

FIG. 18. Fourier transform of the connected density-density correlation function in the steady state, with parameters $L = 500$, $J = 1$ and $\gamma = 3$, in Model I.

clearly visible at small momentum and energy.

### XI.  LINEAR RESPONSE IN LINDBLAD DYNAMICS

The theory of linear response in Lindblad equations was worked out in Ref.[51] and we now briefly review the relevant results. The starting point is a Lindblad equation with Lindbladian $\mathcal{L}_0$. The corresponding time evolution operator for the reduced density matrix $\rho$ is

$$\rho_0(t) = \mathcal{E}_0(t) * \rho , \quad \mathcal{E}_0(t) = e^{\mathcal{L}_0 t} . \tag{159}$$

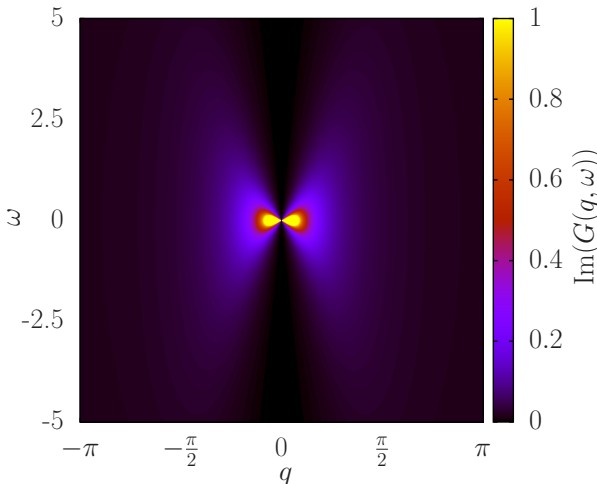

FIG. 19. Fourier transform of the connected density-density correlation function in the steady state, with parameters $L = 500$, $J = 1$ and $\gamma = 3$, in Model II.

One then considers a weak perturbation to the Lindbladian $\mathcal{L}_0$ of the form

$$\mathcal{L}(t) = \mathcal{L}_0 + \xi(t)\mathcal{L}_1 \, , \qquad (160)$$

which induces time-evolution of the density matrix by

$$\rho(t) = \mathcal{E}(t) * \rho \, , \quad \mathcal{E}(t) = T e^{\int_0^t d\tau \, \mathcal{L}(\tau)} \, . \qquad (161)$$

The expectation value of an observable $A$ can then be expanded in powers of the small parameter $\xi$

$$\text{Tr}\left[A\rho(t)\right] - \text{Tr}\left[A\rho_0(t)\right] = \int_0^\infty dt' \xi(t')\chi_{A\mathcal{L}_1}(t,t')$$
$$+ \text{ higher order in } \xi \, , \quad (162)$$

where the linear susceptibility is given by

$$\chi_{A\mathcal{L}_1}(t,t') = \theta(t-t')\text{Tr}\left(A\big(\mathcal{E}_0(t-t')\mathcal{L}_1\mathcal{E}_0(t')\big) * \rho\right) \, . \qquad (163)$$

Focusing on perturbations of the form

$$\mathcal{L}_1[\rho] = -i[B, \rho] \, , \qquad (164)$$

and moving to the Heisenberg picture this can be written as

$$\chi_{AB}(t,t') = i\theta(t-t')\text{Tr}\left(\rho\mathcal{E}_0^*(t') * \big[B, [A(t-t')]\big]\right) . \qquad (165)$$

Here the time evolution of operators is obtained by acting with the dual map

$$A(t) = \mathcal{E}_0^*(t) * A \, . \qquad (166)$$

In general this is quite a complicated object, but for the Lindblad equations considered in our work it simplifies significantly as we will see. In the following we will focus on the density response to a density perturbation, i.e.

$$B = n_\ell \, , \quad A = n_j \, , \qquad (167)$$

where $n_j$ is the number operator for spinless fermions on site $j$. The corresponding dynamical susceptibility is then denoted by

$$\chi(j,\ell;t,t') = i\theta(t-t')\text{Tr}\left(\rho\mathcal{E}_0^*(t') * \big[n_\ell, [n_j(t-t')]\big]\right) . \qquad (168)$$

### A. Linear response in the steady state

The steady state of the unperturbed Limdblad evolution fulfils

$$\mathcal{E}_0(t) * \rho_{\text{SS}} = \rho_{\text{SS}} \, . \qquad (169)$$

The linear response in the steady state takes a particularly simple form [51]

$$\chi_{AB}^{\text{SS}}(t) = i\theta(t)\text{Tr}\left(\rho_{\text{SS}}\big[B, A(t)\big]\right) . \qquad (170)$$

Given that the steady state density matrix is equal to the identity (or the identity in a sector with fixed particle number), density response functions (170) in the steady state vanish.

### B. Linear response functions after a quantum quench

We now focus on the particular case of density-density response functions after a quantum quench (168). These are of interest e.g. for pump-probe experiments. As our initial state we take a Gaussian fermionic state $\rho_0$ characterized by the correlation matrix (all other two-point functions vanish)

$$C_{j,\ell} = \text{Tr}\left[\rho_0 c_j^\dagger c_\ell\right] = f(j-\ell) + (-1)^\ell g(j-\ell) \, . \qquad (171)$$

An example is the ground state of a tight-binding model with dimerization [67], in which case we have

$$f(j) = \frac{1}{L}\sum_{k>0} e^{ikj}\frac{(-1)^j + h^2(k)}{1 + h^2(k)},$$

$$g(j) = \frac{1}{L}\sum_{k>0} e^{ikj}\frac{ih(k)}{1 + h^2(k)}\big((-1)^j - 1\big) \, ,$$

$$h(k) = \frac{2J\cos k - \epsilon_-(k)}{2J\delta\sin k} \, ,$$

$$\epsilon_-(k) = -2J\sqrt{\delta^2 + (1-\delta^2)\cos^2 k} \, . \qquad (172)$$

It is convenient to consider the following Fourier transform of the linear response function (168)

$$\chi(Q, Q', \omega, t') = \frac{1}{L}\sum_{j,\ell}\int_{t'}^\infty dt \, e^{i\omega(t-t')+iQj+iQ'\ell}$$
$$\times \chi(j,\ell;t,t') \, . \qquad (173)$$

Due to broken translational invariance in the initial state there are two contributions

$$\chi(Q, Q', \omega, t') = \delta_{Q,-Q'} \, \chi_{\text{sm}}(Q, \omega, t') \\ + \delta_{Q,\pi-Q'} \, \chi_{\text{st}}(Q, \omega, t') \; . \quad (174)$$

As $\chi_{\text{sm}}(Q, \omega, t')$ is independent of $t'$ in absence of dissipation we focus on $\chi_{\text{st}}(Q, \omega, t')$. For $\gamma = 0$, i.e. purely unitary time evolution, we have for all our models

$$\chi_{\text{st}}^0(Q, \omega, t') = \frac{1}{L} \sum_k \frac{[\tilde{g}(-k) - \tilde{g}(k)]e^{2i\epsilon(k)t'}}{\omega + i\delta + \epsilon(k) - \epsilon(k+Q)}, \quad (175)$$

where $\delta$ is a positive infinitesimal and

$$\tilde{g}(k) = \sum_j e^{ijk} g(j) \; . \quad (176)$$

We stress that even in absence of dissipation, i.e. $\gamma = 0$, the staggered dynamical suceptibilities $\chi_{\text{st}}^0(Q, \omega, t')$ vanish in the late time limit as a result of the restoration of translational invariance [52]. In Fig. 20 we show the real part of $\chi_{\text{st}}^0(Q, \omega, t' = 0)$ as a function of $\omega$ and $Q$. The main features are particle-hole scattering continua,

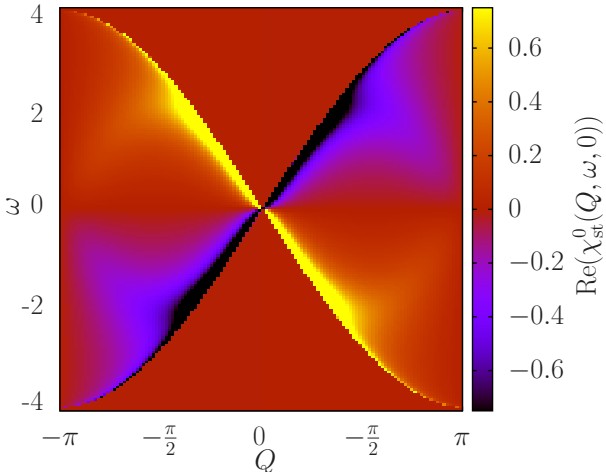

FIG. 20. Real part of $\chi_{\text{st}}^0(Q, \omega, t' = 0)$ for dimerization $\delta = 0.25$.

which exhibit square root singularities when

$$\omega \to \pm 4J \sin(Q/2) \; . \quad (177)$$

This behaviour can be straightforwardly understood from (175) by first turning the sum into an integral in the $L \to \infty$ limit and then using that

$$\text{Im} \frac{1}{\omega + i\delta + z(k)} = -\pi\delta(\omega + z(k)) \; . \quad (178)$$

to extract the most singular part. In Fig. 21 we show the real part of $\chi_{\text{st}}^0(Q, \omega, t' = 2)$ as a function of $\omega$ and $Q$. We observe that the magntiude of $\chi_{\text{st}}^0$ diminishes with

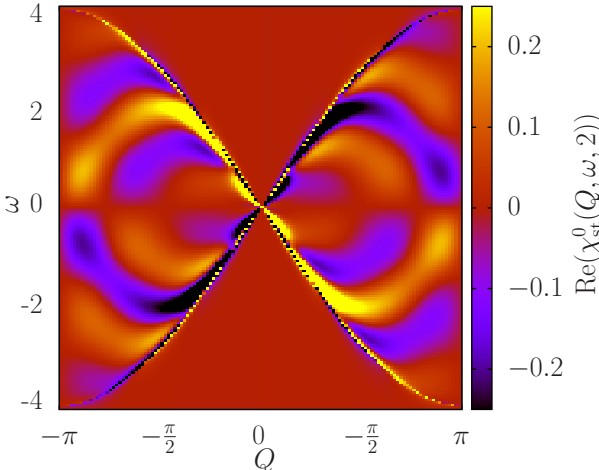

FIG. 21. Real part of $\chi_{\text{st}}^0(Q, \omega, t' = 2)$ for dimerization $\delta = 0.25$.

increasing $t'$, in agreement with the fact that at late $t'$ translational symmetry gets restored. We also observe an "interference pattern" inside the particle-hole continuum.

We now turn to the effect of non-vanishing dissipation on density-density response functions. In Fig. 22 we show the real part of $\chi_{\text{st}}(Q, \omega, t' = 0)$ in Model I for dimerization $\delta = 0.25$ and dissipation rate $\gamma = 0.5$. A comparison

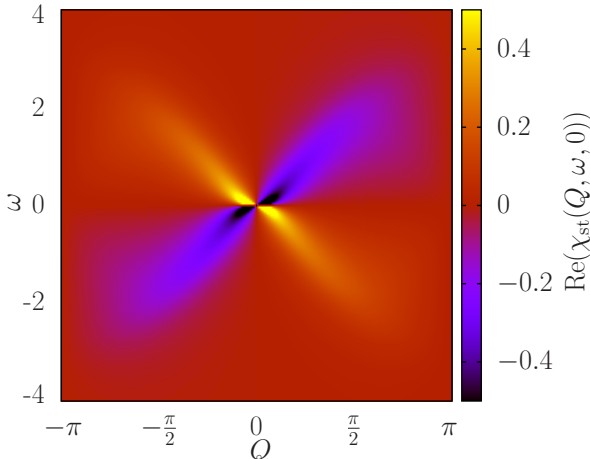

FIG. 22. Real part of $\chi_{\text{st}}(Q, \omega, t' = 0)$ in Model I for dimerization $\delta = 0.25$ and dissipation rate $\gamma = 0.5$.

with Fig. 20 reveals the effects of a non-zero dephasing rate: sharp features like the aforementioned square root singularities visible in absence of dissipation get washed out. These effects are compounded by the time evolution of the density matrix itself, i.e. considering the response function for $t' > 0$. In Fig. 23 we show the real part of $\chi_{\text{st}}(Q, \omega, t' = 2)$ for $L = 500$, dissipation rate $\gamma = 0.1$ and dimerization $\delta = 0.25$. Comparing this to Fig 21 we

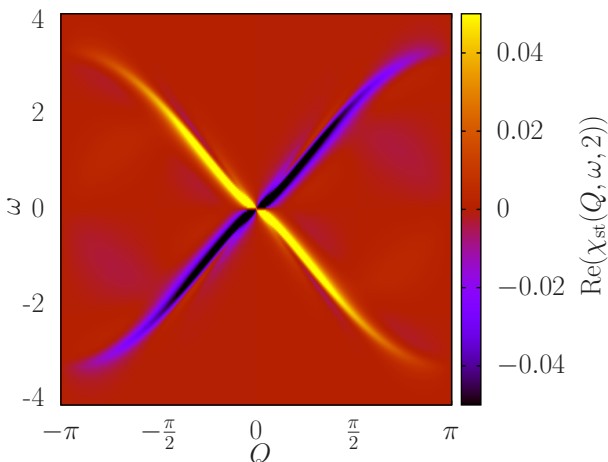

FIG. 23. Real part of $\chi_{\mathrm{st}}(Q, \omega, t' = 2)$ for $L = 500$, dissipation rate $\gamma = 0.1$ and dimerization $\delta = 0.25$.

see that dissipation has essentially erased the modulations inside the particle-hole continuum that are clearly visible in the $\gamma = 0$ evolution.

In order to get a clearer view of how dissipation affects the lineshapes observed in the linear response under unitary time evolution we show some constant momentum cuts in Fig. 24.

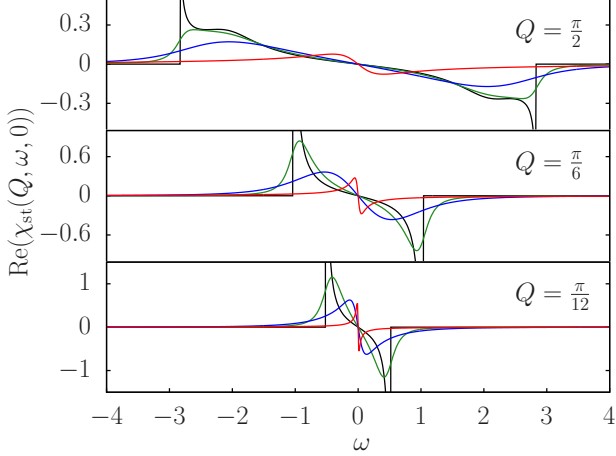

FIG. 24. Constant momentum cuts of the real part of $\chi_{\mathrm{st}}(Q, \omega, t' = 0)$ at different Q values and dissipation rates $\gamma = 0$ (black), $\gamma = 0.1$ (green), $\gamma = 0.5$ (blue), and $\gamma = 5$ (red).

We observe that, as expected, the square root singularities in $\chi_{\mathrm{st}}^0(Q, \omega, t' = 0)$ are smoothed out by even a small dissipation rate $\gamma$. For smaller values of $Q$ signatures of the diffusive mode present in the Lindbladian evolution begin to emerge at low frequencies and larger dissipation rates. This can be seen by substituting the hydrodynamic projection (109) for $n_j(t)$ into the expres-

sion for the linear response function and comparing this to the full result.

The linear response functions in Models II-IV are qualitatively similar to the ones in Model I, and we therefore refrain from presenting results for them.

## XII. SUMMARY AND CONCLUSIONS

In this work we have considered a class of LEs for spinless fermions, which have the property that their respective BBGKY hierarchies decouple. In cases with a U(1) symmetry related to particle number conservation this leads to linear equations of motion for operators involving a fixed number of fermion creation and annihilation operators. Importantly, these are not reducible to the equations of motion for fermion bilinears. We have shown that this makes it possible to obtain exact results for the Heisenberg picture time evolution of operators quadratic in fermions, irrespective of whether the LEs are Yang-Baxter integrable or not. In particular, we have obtained closed-form expressions for the exact hydrodynamic projections of such operators at late times. The relatively simple structure of the Heisenberg equations of motion also makes it possible to determine linear response functions out of equilibrium in the framework of LEs. We have determined the density-density response functions after a quantum quench initialized in the ground state of a tight-binding chain with dimerized hopping. The main effects of dissipation are to wash out sharp features in the lineshapes of dynamical response functions, and to generate signatures of the diffusive mode of the Lindbladian that drives the relaxation towards the steady state at late times. Our work also establishes an interesting new perspective regarding the question how Yang-Baxter integrability affects Lindbladian dynamics, given that the latter converges to the same steady state irrespective of the initial conditions. As we have shown, the Heisenberg-picture dynamics of operators in our LEs can be mapped onto few-particle imaginary time Schrödinger equations with non-Hermitian Hamiltonians. Integrability then imposes simple forms of the eigenfunctions of these Hamiltonians.

Our work opens the door to several lines of enquiry. First, the decoupling of the BBGKY hierarchy holds on any lattice in any number of dimensions. It would be interesting to investigate whether explicit results e.g. for hydrodynamic projections can be obtained in higher dimensions. Second, we believe that the simpler structure of eigenstates of the aforementioned non-Hermitian Hamiltonians should lead to qualitative differences between integrable and non-integrable LEs in appropriated chosen complexity measures of time-evolved operators, cf. [68] for related work. This question is currently under investigation. Finally, it would be interesting to consider the effects of dissipative boundaries that preserve the truncation of the BBGKY hierarchy [69–74]. This would enable the study of transport properties involving

two-particle Green's functions.

## ACKNOWLEDGEMENTS

This work was supported in part by the EPSRC under grant EP/X030881/1. We are grateful to Denis Bernard, Thierry Giamarchi and Curt von Keyserlingk for stimulating discussions.

## Appendix A: Diagonalizing the dual Lindbladian in momentum space

When carrying out numerical computations it is sometimes convenient to consider the equations of motion in momentum space. We use conventions such that

$$c_j = \frac{1}{\sqrt{L}} \sum_q e^{-iqj} c(q) \, , \ q = \frac{2\pi n}{L} \, , \ -\frac{L}{2} \leq n < \frac{L}{2} \, , \quad \text{(A1)}$$

and consider the following set of operators

$$\mathcal{O}_{n,m}(\mathbf{k}) = c^\dagger(k_1) \ldots c^\dagger(k_n) c(k_{n+1}) \ldots c(k_{n+m}) \, . \quad \text{(A2)}$$

These carry non-zero momentum

$$Q = \sum_{j=1}^n k_j - \sum_{j=1}^m k_{n+j} \, . \quad \text{(A3)}$$

The dual Lindbladian acts as

$$\mathcal{L}^*[\mathcal{O}_{n,m}(\mathbf{k})] = \left( \sum_{j=1}^n \epsilon^*(k_j) + \sum_{j=1}^m \epsilon(k_{n+j}) \right) \mathcal{O}_{n,m}(\mathbf{k}) + \frac{2\gamma}{L} \sum_{\alpha < \beta = 1}^{n+m} \sum_p S_{(\alpha,\beta)}(p)[\mathcal{O}_{n,m}(\mathbf{k})] \, , \quad \text{(A4)}$$

where

$$\epsilon(q) = 2iJ\cos(q) - \gamma \, , \quad \text{(A5)}$$

and $S_{(\alpha,\beta)}(p)$ are model-dependent, momentum conserving kernels acting in the following way

$$\text{I:} \ S_{(\alpha,\beta)}(p)[\mathcal{O}_{n,m}(\mathbf{k})] = \begin{cases} \mathcal{O}_{n,m}\big(\mathbf{k} + p(\mathbf{e}_\alpha + \mathbf{e}_\beta)\big) & \text{if } \alpha \leq n < \beta \, , \\ 0 & \text{else} \, , \end{cases} \quad \text{(A6)}$$

$$\text{II:} \ S_{(\alpha,\beta)}(p)[\mathcal{O}_{n,m}(\mathbf{k})] = \begin{cases} [\cos(\mathbf{k}(\mathbf{e}_\alpha - \mathbf{e}_\beta))]\mathcal{O}_{n,m}\big(\mathbf{k} + p(\mathbf{e}_\alpha + \mathbf{e}_\beta)\big) & \text{if } \alpha \leq n < \beta \, , \\ \cos(p) \, \mathcal{O}_{n,m}\big(\mathbf{k} + p(\mathbf{e}_\alpha - \mathbf{e}_\beta)\big) & \text{else} \, , \end{cases} \quad \text{(A7)}$$

$$\text{III:} \ S_{(\alpha,\beta)}(p)[\mathcal{O}_{n,m}(\mathbf{k})] = \begin{cases} [1 + \nu\cos(p)]\mathcal{O}_{n,m}\big(\mathbf{k} + p(\mathbf{e}_\alpha + \mathbf{e}_\beta)\big) & \text{if } \alpha \leq n < \beta \, , \\ \cos(p) \, \mathcal{O}_{n,m}\big(\mathbf{k} + p(\mathbf{e}_\alpha - \mathbf{e}_\beta)\big) & \text{else} \, , \end{cases} \quad \text{(A8)}$$

$$\text{IV:} \ S_{(\alpha,\beta)}(p)[\mathcal{O}_{n,m}(\mathbf{k})] = \begin{cases} [\cos(\mathbf{k}(\mathbf{e}_\alpha - \mathbf{e}_\beta)) + \xi\cos(\mathbf{k}(\mathbf{e}_\alpha + \mathbf{e}_\beta) + p)]\mathcal{O}_{n,m}\big(\mathbf{k} + p(\mathbf{e}_\alpha + \mathbf{e}_\beta)\big) & \text{if } \alpha \leq n < \beta \, , \\ \cos(p) \, \mathcal{O}_{n,m}\big(\mathbf{k} + p(\mathbf{e}_\alpha - \mathbf{e}_\beta)\big) & \text{else} \, , \end{cases} \quad \text{(A9)}$$

$$\text{V:} \ S_{(\alpha,\beta)}(p)[\mathcal{O}_{n,m}(\mathbf{k})] = \begin{cases} (\cos(\mathbf{k}(\mathbf{e}_\alpha - \mathbf{e}_\beta)) - \cos(p)) \, T_{\alpha,\beta}[\mathcal{O}_{n,m}\big(\mathbf{k} + p(\mathbf{e}_\alpha + \mathbf{e}_\beta)\big)] & \text{if } \alpha \leq n < \beta \, , \\ 0 & \text{else} \, . \end{cases} \quad \text{(A10)}$$

Here $\mathbf{e}_\alpha$ are unit vectors with a single entry at position $\alpha$. Finally, for Model V we have introduced operators $T_{\alpha,\beta}$ that act on $\mathcal{O}_{n,m}$ as

$$T_{\alpha,\beta}[\mathcal{O}_{n,m}(\mathbf{k})] = c^\dagger(k_1) \ldots c^\dagger(k_{\alpha-1}) c(k_\beta) c^\dagger(k_{\alpha+1}) \ldots c^\dagger(k_n) c(k_{n+1}) \ldots c(k_{\beta-1}) c^\dagger(k_\alpha) c(k_{\beta+1}) \ldots c(k_{n+m}) \, . \quad \text{(A11)}$$

"Normal-ordering" this expression such that all creation operators are on the right generates contributions involv-

ing $\mathcal{O}_{n-1,m-1}$ and $\mathcal{O}_{n-2,m-2}$.

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
