# Peer review of "Linear response and exact hydrodynamic projections in Lindblad equations with decoupled Bogoliubov hierarchies"

_SciPost Physics_

## Round 1 · Referee Report · Anonymous (Referee 1) · 2025-10-18

Strengths

1- Very. through and detailed 2- Authors go beyond hydrodynamic projection and study linear response

Weaknesses

None identfied

Report

The authors study hydrodynamic projections in Lindblad master equations of spinless fermions that have a U(1) $\em{strong}$ [New J. Phys. 14 073007 (2012)] symmetry (models I-IV) or do not (model V). The method used is the BBGKY hierarchy, which decouples and allows mapping of the dynamics of operators onto few body operators (using a non-Hermitian Hamiltonian in a doubled Hilbert space aka "vectorization"). The authors obtain exact hydrodynamic projections of quadratic in fermion operators and linear response functions. To elaborate a bit more, hydrodynamic projection is a method that allows one to find the late time power-law behaviour of operators related to a conservation law by projecting onto e.g. quadratically extensive charges for diffusion. To emphasize, the BBGKY decoupling is unrelated to integrability.

I find this paper insightful and well written. It will certainly contribute to the literature on Lindbladian hydrodynamics. I strongly recommend publication in SciPost Physics.

Requested changes

I only would like to ask the authors to consider the following suggested optional changes and questions:

  1. In the introduction (p.2, lhs column): "The presence of a conservation law then implies that certain observables will exhibit hydrodynamic power-law tails at late times, which are related to the existence of diffusive eigenmodes of the Lindbladian" - Is this obvious? Maybe one would need to have add certain caveats?

  2. "In contrast to unitary quench dynamics [50] integrability in LEs does not imply the existence of conservation laws. This is immediately obvious from the fact that many integrable LEs have unique steady states that are completely mixed, i.e. correspond to infinite temperature density matrices." - one should cite the relevant papers e.g. in [21-31]. For instance, in [26] the unique steady state is all down (up to certain boundary bound states and the formation of domain walls in the thermodynamic limit).

  3. p. 7, lhs column: In Fig 3 why are there (two) gaps? Is there a physical reason? In fact Fig. 3 does not seem to be referenced in the text.

  4. p. 10, rhs column: "B. Spatially local operators" - why spatially local? It seems to me to be more like "Spatially local without well defined momentum"? Maybe renaming this would make things clearer.

  5. In Sec. VI dealing with quadratic operators in model V, it would be possibly interesting to see how the operators exponentially decay? Or is it not interesting?

  6. p. 17, lhs column: "It would be interesting to pursue a more precise description by treating the ”particles” as interacting and following the logic of Ref. [64]." - This claim I find interesting and would ask that the authors elaborate slightly.

  7. Is the relevance of momentum $\pi$ for the eigenvalues closing to 0 related to a possible $\eta$-pairing operator (at momentum $\pi$) that appears from "spin up and down" vectorized fermions? Can the authors comment?

Recommendation

Publish (surpasses expectations and criteria for this Journal; among top 10%)

---

## Round 1 · Referee Report · Anonymous (Referee 2) · 2025-12-6

Strengths

1- Presentation of an innovative approach to calculate the evolution of time-dependent correlation functions in dissipative, non-Gaussian systems 2- The technique presented in this work clearly opens a new pathway to investigate the non-trivial dynamics of a broad class of physically important dissipative systems 3- Even the most involved calculations are clearly presented and easy to follow 4- The technique presented in this work is of broad interested for the theoretical community interested in the dynamics of correlated quantum systems

Weaknesses

1- Even though the paper is very well written, it seems to be mainly addressed to the scientific community comfortable with jargon and concepts currently used to describe the hydrodynamic behavior of one-dimensional dissipative and integrable systems. I think that the techniques and results presented in this paper are of much broader scientific interest. Some additional paragraphs of explanation and context would make the paper accessible to a broader community, as it deserves. 2- There is some lack of physical discussion of the results and lack of motivation for some of the chosen models

Report

In this work, the authors build upon a correlation-to-wave-function correspondence first devised in Medvedyeva, Essler, Prosen, PRL 117, 137202 (2016) (Ref. [21] in the submitted manuscript) to provide exact analytical results concerning the time-evolution of n-body operators and linear-response functions in a broad class of dissipative models.

The important fact is that such models are governed by non-Gaussian Lindblad equations, which are notoriously important for correlated out-of-equilibrium physics and difficult to solve. Specifically, such models describe the Lindblad evolution of free fermions under the action of Hermitian and quadratic jump operators. Even though such models are non-Gaussian, a double commutator structure does not generate higher-order correlation functions, leading to a close set of equations of motions for the operators or, equivalently, decoupling the BBGKY hierarchy. The authors show that a well-suited vectorization procedure maps these equations of motion onto effective few-body problems (two-body for quadratic operators, four-body for quartic operators, ...). The authors provide exact solutions for the two-body problem in various models, even without particle conservation. In the specific case of dephasing processes, a mapping over a complex version of the spinful Hubbard model in one-dimension, allows to rely on a Bethe-Ansatz solution to extend results beyond quadratic operators.

The authors provide a nice demonstration of the effectiveness and importance of their approach, by deriving exactly the long-time hydrodynamic and diffusive behavior of quadratic operators associated to conserved charges (such as the particle number). This is equivalent to finding the so-called "hydrodynamic projections" of such operators, which is an important result for the quantum transport community. Furthermore, the authors ingeniously extend their approach to exactly derive correlation functions between quadratic operators at different times, notoriously important to study dynamical linear responses. They focus on the case of density-density response functions, normally probed in quantum transport and pump-and-probe experiments, and illustrate some effects of the dissipation.

The methods and results illustrated in this work are of great interest for the broad physics community interested in the quantum transport properties of correlated and dissipative systems. For this reason, I strongly recommend publication in SciPost Phyisics. However, as mentioned in the "Weakenesses" section, I think this work deserves to be made more accessible to a broader community by further explaining some concepts and jargon, including also some additional motivation and explanation concerning the physical systems and phenomena which could be explored based on the developed approach (see requested changes below).

Requested changes

[Not ranked in order of importance, but in order of my annotations to the text]

1- After Eq. (2): "Assuming the jump operators are Hermitian, L†α = Lα , or come in adjoint pairs, L†α = Lβ". Definitively correct. However, the last case falls back on the Hermitian one when we redefine couples of jump operators (Lα + L†α)/\sqrt2 and (Lα - L†α)/(i\sqrt2), leading to the same Lindblad equation. Maybe an additional word to clarify this aspect may be useful. 2- As the authors state, Models III and IV are elaborations of Models I and II respectively. Is there a specific reason why the authors chose these elaborations ?
3- Eq. (14), should the label "r" be replaced by the label "s" in the very last exponent of (-1)? 4- After Eq. (19) the authors state: "A key feature of this Hamiltonian is that in all terms there are at least as many fermion annihilation operators as there are creation operators." Maybe I am missing a key point here, but this does not seem to be the case for the last two lines of Eq. (19). This further conflicts with a statement in page 13: "The vectorized Hamiltonian (19) has the key property that its interaction terms in H contain more annihilation that creation operators" 5-"more annihilation that creation" -> "more annihilation thaN creation"?
6- After equation (22): "non-positive" -> "negative" would simplify the reading
7- What about the \sigma index for n_j in the last line of Eq. (19)? 8- Title of Section IV.A : n_\sigma = \sum_j n_j\sigma ? For pedagogical reasons, stating explicitly, before Section IV.A, that you map the equations of motion for quadratic operators onto a 2-body problem may be helpful to those getting acquainted with the formalism. 9- Eq. (28). If I get it right, here the \sigma label becomes suddenly a space label, while it was associated to the pseudo-spin in the previous section. For clarity, it would be better to change label. 10- Page 6: "This reflects the existence of a diffusive hydrodynamic mode related to the particle number U (1) symmetry of the model." Of course I agree, but this could be quite obscure to the general reader. 11- After Eq. (47) the authors point to the relation between two-particle bound states and diffusive modes. Can the authors elaborate more on such connection? What do we learn from it ? 12- Section IV.B for models with n_\uparrow=2. As the authors state, it is totally expected that the charged operators decay exponentially. Is there something then that we should learn from the exact calculation of the exponential decay, beyond its analytic expression?
13- Section V. If I understood correctly, the authors rely on their wave-function formalism to calculate how quadratic operators decompose on the longest lived eigen-modes of the Lindblad 2-body operator sector. They can so analytically derive their diffusive behavior. According to their terminology, this procedure unveils "the hydrodynamic projection" of such operators on the diffusive modes. This is a very interesting result! However, I had to go through the entire calculation to (re)construct this line of reasoning. Maybe illustrating the philosophy of the calculation with more than a "We now show how to explicitly evaluate (70) for Model I", before starting developing the technicalities (maybe even in the introductory Section I), may help the reader following why some calculations are done. The sentence "this is a key result of our work" after Eq. (91) comes out of the blue and it is not really said why. I had to construct the (very interesting indeed!) answer myself looking at the following equations and plots. 14- Fig. 10. Why exactly does the time evolution of the circles corresponding to (q,d)=(\pi/5,10) initially follow the curve of (0,1) and suddenly change slope ? 15- Section VI. Study of Model V. This is a quite interesting model because it is the only one without charge conservation, but still solvable as decoupling the BBGKY hierarchy. Nonetheless, the authors derive exclusively the 2-body spectrum associated to quadratic operators and derive their expected exponential decay, which is expected. Could the authors briefly elaborate on some more interesting physical effects which could be studied in this model thanks to their analytical solution? 16-The appearance in Fig. 17 of the 2-particle continuum in the Lindblad spectrum is extremely intriguing. Could be this understood based on an analogue of the Lehmann representation of the response function extended to the Lindblad formalism? 17- "simnple" -> "simple" before Eq. (135) 18- Is the correct interpretation of Eq. (135) that Eq. (134) can be understood as the scalar product of the two eigen-functions corresponding to the density operators at time t and time zero ? 19- Conclusions. "This would enable the study of transport properties involving two-particle Green’s functions [67, 68]". This research program has been partially already addressed (to my knowledge) in https://iopscience.iop.org/article/10.1088/1742-5468/2010/05/L05002 https://journals.aps.org/prb/abstract/10.1103/PhysRevB.104.144301 https://journals.aps.org/prresearch/abstract/10.1103/PhysRevResearch.4.013109 https://journals.aps.org/prl/abstract/10.1103/PhysRevLett.132.136301 It may be also important to mention that an alternative self-consistent method to calculate two-point correlation functions was also devised in https://journals.aps.org/prb/abstract/10.1103/PhysRevB.102.100301 20- I could not find any reference to the Appendix in the main text

Recommendation

Publish (easily meets expectations and criteria for this Journal; among top 50%)

---

## Round 2 · Author Response

Reply to the first referee
We are grateful to the referee for their positive assessment of our manuscript and their helpful comments. We have revised the manuscript in order to address them. In the following we respond to the individual points raised by the refereee.
-
In the introduction (p.2, lhs column): "The presence of a conservation law then implies that certain observables will exhibit hydrodynamic power-law tails at late times, which are related to the existence of diffusive eigenmodes of the Lindbladian" - Is this obvious? Maybe one would need to have add certain caveats?
- In our revised manuscript, we have included in the introduction a discussion of the physical argument that relates the presence of a conservation law to hydrodynamic power-law tails at late times (for inhomogeneous initial states). Employing a spectral representation in terms of left and right eigenstates of the Lindbladian then would seem to imply "generically" the existence of diffusive eigenmodes. All of these are plausible arguments rather than proofs, but we are not aware of counterexamples.
-
"In contrast to unitary quench dynamics [50] integrability in LEs does not imply the existence of conservation laws. This is immediately obvious from the fact that many integrable LEs have unique steady states that are completely mixed, i.e. correspond to infinite temperature density matrices." - one should cite the relevant papers e.g. in [21-31]. For instance, in [26] the unique steady state is all down (up to certain boundary bound states and the formation of domain walls in the thermodynamic limit).
- We thank the referee for this suggestion, which we have implemented.
-
p. 7, lhs column: In Fig 3 why are there (two) gaps? Is there a physical reason? In fact Fig. 3 does not seem to be referenced in the text.
- We thank the referee for pointing out the missing reference. This has now been fixed. Regarding the two "branches" of the dispersion: these arise because the solution only exists in a particular range of momenta as is mentioned after the expression for $\eta$. The reason for this is the same as for bound state solutions in Hamiltonian many-particle systems: bound states exist only in momentum regions where the effective interactions of the constituent "quasi-particles" are attractive.
-
p. 10, rhs column: "B. Spatially local operators" - why spatially local? It seems to me to be more like "Spatially local without well defined momentum"? Maybe renaming this would make things clearer.
- "Spatially local" refers to operators that act on a finite spatial region in the thermodynamic limit. This is a widely used terminology in the literature, see e.g. Ref.[52] of the revised version of the manuscript. The operators carrying a definite momentum are not spatially local in this sense (only by smearing out the momentum could one impose a form of spatial locality, which however is weaker than the one of $c^\dagger_{x_0}c_{y_0}$).
-
In Sec. VI dealing with quadratic operators in model V, it would be possibly interesting to see how the operators exponentially decay? Or is it not interesting?
- We thank the referee for this suggestion. It would indeed be interesting, but given that our manuscript is already very long and contains a large number of new results we are concerned that this would simply be lost. We have added a remark at the end of the section setting out what kind of physical problem such a calculation would be able to address. "A physical process that could be analyzed in model V would be the time evolution of particle density and its fluctuations in a quantum quench starting from an initially empty state."
-
p. 17, lhs column: "It would be interesting to pursue a more precise description by treating the ”particles” as interacting and following the logic of Ref. [64]." - This claim I find interesting and would ask that the authors elaborate slightly.
- We have extended this comment following the referee's suggestion.
-
Is the relevance of momentum $\pi$ for the eigenvalues closing to 0 related to a possible $\eta$-pairing operator (at momentum $\pi$) that appears from "spin up and down" vectorized fermions? Can the authors comment?
- For model I the existence of an $\eta$-pairing structure was discussed in some detail in Ref.[21] of the revised manuscript. For the other models the relevance of the gap closing at momentum $\pi$ is a consequence of our vectorization procedure, which is constructed to generate the same sign of the kinetic terms in the Hamiltonians for spin-up and spin-down fermions. This induces a momentum shift by $\pi$. In other words, the gap closing at $\pi$ in the vectorized picture corresponds to gap closing at momentum zero in the unvectorized picture, which is precisely what one expect for diffusive behaviour.
Reply to the second referee
We are grateful to the referee for their positive assessment of our manuscript and their many helpful comments. We have revised the manuscript substantially in order to take them into account. In the following we respond to the individual points raised by the refereee.
-
"After Eq. (2): Assuming the jump operators are Hermitian, $L^\dagger_\alpha=L_\alpha$, or come in adjoint pairs, $L^\dagger_\alpha=L_\beta$". Definitively correct. However, the last case falls back on the Hermitian one when we redefine couples of jump operators $(L_\alpha + L^\dagger_\alpha)/\sqrt{2}$ and $(L_\alpha - L^\dagger_\alpha)/(i\sqrt{2})$, leading to the same Lindblad equation. Maybe an additional word to clarify this aspect may be useful.
- We thank the referee for this comment. We would like to keep the discussion short and simple and therefore have rephrased the sentence to "If the set ${L_\alpha}$ of jump operators is closed under Hermitian conjugation,...".
-
As the authors state, Models III and IV are elaborations of Models I and II respectively. Is there a specific reason why the authors chose these elaborations ?
- Models III and IV are chosen in order to give a clear picture of how the spatial structure of the jump operators affects the properties of the hydrodynamic modes, without making the analysis overly tedious. We have added a remark explaining this at the end of section II.
-
Eq. (14), should the label "r" be replaced by the label "s" in the very last exponent of (-1)?
- We thank the referee for spotting this typo. We have corrected it.
-
*After Eq. (19) the authors state: "A key feature of this Hamiltonian is that in all terms there are at least as many fermion annihilation operators as there are creation operators." Maybe I am missing a key point here, but this does not seem to be the case for the last two lines of Eq. (19). This further conflicts with a statement in page 13: "The vectorized Hamiltonian (19) has the key property that its interaction terms in H contain more annihilation that creation operators" *
- We are unsure what the referee has in mind here. The last two lines of eqn (19) in the old manuscript (eqn (32) in the revised version) contain indeed more annihilation than creation operators, because $P_j$ are pair annihilation operators. Hence the various terms have respectively 2 creation and 2 annihilation operators, 1 creation and 3 annihilation operators, or zero creation and 4 annihilation operators. So the first statement is indeed correct. The second statement the referee quotes on the other hand is indeed misleading, as some of the interaction terms have the same number of creation and annihilation operators. We have corrected this.
-
"more annihilation that creation" $\rightarrow$ "more annihilation thaN creation"?
- We thank the referee for spotting this typo. We have corrected it.
-
After equation (22): "non-positive" $\rightarrow$ "negative" would simplify the reading
- We chose the wording "non-positive" to indicate that the eigenvalue zero is generally allowed.
-
What about the $\sigma$ index for $n_j$ in the last line of Eq. (19)?
- We thank the referee for spotting this. We have added the definition $n_j=n_{j,\uparrow}+n_{j,\downarrow}$ after the equation.
-
Title of Section IV.A : $n_\sigma = \sum_j n_j\sigma$ ? For pedagogical reasons, stating explicitly, before Section IV.A, that you map the equations of motion for quadratic operators onto a 2-body problem may be helpful to those getting acquainted with the formalism.
- We thank the referee for this suggestion. We have changed the manuscript accordingly.
-
Eq. (28). If I get it right, here the $\sigma$ label becomes suddenly a space label, while it was associated to the pseudo-spin in the previous section. For clarity, it would be better to change label.
- We have followed the referee's suggestion and changed the label from $\sigma$ to $\tau$.
-
Page 6: "This reflects the existence of a diffusive hydrodynamic mode related to the particle number U (1) symmetry of the model." Of course I agree, but this could be quite obscure to the general reader.
- We have added an explanation of this comment.
-
After Eq. (47) the authors point to the relation between two-particle bound states and diffusive modes. Can the authors elaborate more on such connection? What do we learn from it ?
- We thank the referee for this question. In response we have added a discussion of diffusive eigenmodes in a new section "Overview of main results". The bound state nature of the diffusive eigenmodes means that they can be written as sums over particle-hole bilinears that are exponentially localized around their "centre of mass". The effective size of the bound state translates into the size of these particle-hole bilinears.
-
Section IV.B for models with $n_\uparrow=2$. As the authors state, it is totally expected that the charged operators decay exponentially. Is there something then that we should learn from the exact calculation of the exponential decay, beyond its analytic expression?
- This section is included mainly for the sake of completeness. We deal that the explicit expressions for eigenoperators are still interesting and allow for the calculation of the exponential decay of Heisenberg picture operators.
-
Section V. If I understood correctly, the authors rely on their wave-function formalism to calculate how quadratic operators decompose on the longest lived eigen-modes of the Lindblad 2-body operator sector. They can so analytically derive their diffusive behavior. According to their terminology, this procedure unveils "the hydrodynamic projection" of such operators on the diffusive modes. This is a very interesting result! However, I had to go through the entire calculation to (re)construct this line of reasoning. Maybe illustrating the philosophy of the calculation with more than a "We now show how to explicitly evaluate (70) for Model I", before starting developing the technicalities (maybe even in the introductory Section I), may help the reader following why some calculations are done. The sentence "this is a key result of our work" after Eq. (91) comes out of the blue and it is not really said why. I had to construct the (very interesting indeed!) answer myself looking at the following equations and plots.
- We thank the referee for this very helpful comment. In response we have made substantial changes to the manuscript. In particular we have introduced a new section "Overview of main results" just after specifying the class of Lindblad equations we are studying. The new section summarizes our main results and, we hope, will make the rest of the manuscript much easier to digest.
-
Fig. 10. Why exactly does the time evolution of the circles corresponding to $(q,d)=(\pi/5,10)$ initially follow the curve of $(0,1)$ and suddenly change slope ?
- There are two kinds of Lindbladian eigenstates that contribute here. The initial decay rate is set by the two-particle continuum shown in Fig. 1. At late times the decay rate is set by the bound state solution shown in Fig. 2. As we are considering an operator with definite momentum, only a single momentum of the bound state solution will contribute. However, this has a very small overlap with the operator under consideration. This is why this behaviour only becomes visible at late times, when the contributions due to the two-particle continuum have already decayed to negligible values. We have added a discussion of this point in the manuscript.
-
Section VI. Study of Model V. This is a quite interesting model because it is the only one without charge conservation, but still solvable as decoupling the BBGKY hierarchy. Nonetheless, the authors derive exclusively the 2-body spectrum associated to quadratic operators and derive their expected exponential decay, which is expected. Could the authors briefly elaborate on some more interesting physical effects which could be studied in this model thanks to their analytical solution?
- The main purpose of our discussion of the time evolution of fermion bilinears is to demonstrate how the "triangular structure" of the BBGKY hierarchy operates in practice. A physical process that could be analyzed in this model would be the time evolution of particle density and its fluctuations in a quantum quench starting from an initially empty state. We have added a brief comment to this effect.
-
The appearance in Fig. 17 of the 2-particle continuum in the Lindblad spectrum is extremely intriguing. Could be this understood based on an analogue of the Lehmann representation of the response function extended to the Lindblad formalism?
- The existence of a continuum of two bound states is entirely expected on the basis of kinematic considerations in the framework of our analysis. What is however interesting, and non-trivial, is whether and how these bound states interact. We address this question in detail for model I in a forthcoming publication.
-
"simnple" $\rightarrow$ "simple" before Eq. (135)
- We thank the referee for spotting this typo. We have corrected it.
-
Is the correct interpretation of Eq. (135) that Eq. (134) can be understood as the scalar product of the two eigen-functions corresponding to the density operators at time t and time zero ?
- Yes indeed. This is because the steady state is an infinite temperature state, which makes it possible to interpret the expectation value as a scalar product on the space of operators.
-
Conclusions. "This would enable the study of transport properties involving two-particle Green’s functions [67, 68]". This research program has been partially already addressed (to my knowledge) in https://iopscience.iop.org/article/10.1088/1742-468/2010/05/L05002 https://journals.aps.org/prb/abstract/10.1103/PhysRevB.104.144301 https://journals.aps.org/prresearch/abstract/10.1103/PhysRevResearch.4.013109 https://journals.aps.org/prl/abstract/10.1103/PhysRevLett.132.136301 It may be also important to mention that an alternative self-consistent method to calculate two-point correlation functions was also devised in https://journals.aps.org/prb/abstract/10.1103/PhysRevB.102.100301
- We thank the referee for pointing out these references. We have added them all at appropriate places in the manuscript.
-
I could not find any reference to the Appendix in the main text.
- We thank the referee for spotting this omission. We have added the intended reference.

---

## Round 2 · List of Changes

The changes are highlighted in red in the pdf file.
Summary of the main changes:
-
A brief overview of hydrodynamic, diffusive late-time dynamics is added to the introduction.
-
A new section is added with an overview of the main results.
-
In Section IV. A. we give a detailed structure of the eigenstates.
-
At the beginning of Section V. we summarise the steps that will follow.

---

## Editorial Decision

unknown